# Computational mechanisms of curiosity and goal-directed exploration

**Philipp Schwartenbeck**[1,2,3,4]*, **Johannes Passecker**[5,8], **Tobias U Hauser**[1,6], **Thomas HB FitzGerald**[1,6,7], **Martin Kronbichler**[2,3], **Karl J Friston**[1]

[1]Wellcome Centre for Human Neuroimaging, University College London, London, United Kingdom; [2]Centre for Cognitive Neuroscience, University of Salzburg, Salzburg, Austria; [3]Neuroscience Institute, Christian-Doppler-Klinik, Paracelsus Medical University Salzburg, Salzburg, Austria; [4]Oxford Centre for Functional MRI of the Brain, Nuffield Department of Clinical Neurosciences, University of Oxford, Oxford, United Kingdom; [5]Department for Cognitive Neurobiology, Center for Brain Research, Medical University Vienna, Vienna, Austria; [6]Max Planck University College London Centre for Computational Psychiatry and Ageing Research, London, United Kingdom; [7]Department of Psychology, University of East Anglia, Norwich, United Kingdom; [8]Mortimer B. Zuckerman Mind Brain and Behavior Institute, New York, United States

**Abstract** Successful behaviour depends on the right balance between maximising reward and soliciting information about the world. Here, we show how different types of information-gain emerge when casting behaviour as surprise minimisation. We present two distinct mechanisms for goal-directed exploration that express separable profiles of active sampling to reduce uncertainty. 'Hidden state' exploration motivates agents to sample unambiguous observations to accurately infer the (hidden) state of the world. Conversely, 'model parameter' exploration, compels agents to sample outcomes associated with high uncertainty, if they are informative for their representation of the task structure. We illustrate the emergence of these types of information-gain, termed active inference and active learning, and show how these forms of exploration induce distinct patterns of 'Bayes-optimal' behaviour. Our findings provide a computational framework for understanding how distinct levels of uncertainty systematically affect the exploration-exploitation trade-off in decision-making.
DOI: https://doi.org/10.7554/eLife.41703.001

*For correspondence:
pschwartenbeck@gmail.com

**Competing interests:** The authors declare that no competing interests exist.

## Introduction

The balance between *exploitation*, that is choosing the most valuable option given current beliefs about the world, and *exploration*, that is choosing options that allow us to forage and learn about our environment, lies at the heart of decision-making and adaptive behaviour (*Cohen et al., 2007*; *Gottlieb et al., 2013*). The trade-off between choosing to exploit or explore is a key focus of computational theories of behaviour in both artificial intelligence and neuroscience, such as in reinforcement learning and Bayesian models of behaviour (*Friston et al., 2015*; *Friston et al., 2017a*; *Sun et al., 2011*; *Sutton and Barto, 1998a*; *Houthooft et al., 2016*; *Hauser, 2018*). Importantly, recent behavioural evidence suggests that humans perform a mixture of both *random* and *goal-directed* exploration (*Gershman, 2018a*; *Wilson et al., 2014a*). Random exploration has been introduced in early accounts of exploratory behaviour (*Daw et al., 2006*; *Sutton and Barto, 1998a*). This behaviour is defined as a deviation from the currently most valuable policy by randomly sampling any other option. A classical way of formalising random exploration is via $\epsilon$-greedy or softmax choice

rules, where in the latter the tendency towards randomness is governed by an inverse temperature parameter (*Sutton and Barto, 1998a*). A more refined account of random exploration has been introduced via Thompson sampling (*Thompson, 1933*), where an agent samples from a posterior over reward statistics and chooses the most valuable option with respect to this sample, thus taking its uncertainty over reward statistics into account (*Agrawal and Goyal, 2011*; *Speekenbrink and Konstantinidis, 2015*).

In contrast to random exploration, goal-directed, information-seeking exploration is guided by the uncertainty in an agent's model of the structure of the world. This implies that agents will selectively sample options that are informative, that is that are associated with the highest uncertainty. A prominent example of uncertainty-sensitive exploration is the upper confidence bound algorithm (*Agrawal, 1995*; *Auer et al., 2002*; *Kaelbling, 1994*; *Sutton and Barto, 1998a*), which adds an uncertainty bonus (*Kakade and Dayan, 2002*) to options that have not been sampled for a long time or that are associated with high uncertainty. See (*Gershman, 2018a*; *Gershman, 2018b*) for a discussion of these two types of exploration and specific predictions arising from these formulations.

It is challenging to provide a formal account of the trade-off between behaviour that aims at maximising reward and fulfils an agent's preferences over states on the one hand, and acquiring information about the world on the other. Furthermore, an important challenge lies in moving beyond descriptive accounts of behaviour towards understanding the generative mechanisms of information gain that could be implemented by a biological system. A particularly challenging aspect lies in providing a formal account of goal-directed exploration, where agents are 'intrinsically motivated' to minimise uncertainty and actively learn about the world, closely linked to the concept of curiosity (*Kidd and Hayden, 2015*; *Oudeyer and Kaplan, 2007*). This is particularly delicate because one can dissociate different types of uncertainties. For example, if an agent is offered an option that may have a positive or a negative outcome, she will be in a state of uncertainty at two levels. First, she has no idea about the probabilities of winning or losing. For example, there could be a 50% or 99% chance of winning. Second, even if she knew the probability of winning exactly (e.g. 50%), there will still be some uncertainty about the outcome if she chose the option (whether she wins or not). These types of uncertainties have been termed unexpected and expected uncertainty (*Yu and Dayan, 2005*) or, in economics, ambiguity and risk. The key point is that it is necessary to resolve ambiguity first before agents can assess the value of options and their associated risk.

We discuss these different aspects of uncertainty-reduction in terms of Bayesian inference, by casting choice behaviour and planning as variational probabilistic inference (*Friston et al., 2013*; *Friston et al., 2017a*). Here, agents are assumed to form expectations over observable states (outcomes) and infer policies that minimise the expected information-theoretic surprise about these observations. These expectations reflect an agent's preferences over observations, such that undesired outcomes will be (a priori) unexpected and surprising. Thus, by minimising surprise, agents find policies that make visiting preferred states more likely. This information-theoretic quantity can be approximated by the expected free energy, which is a function of (approximate posterior) beliefs about the states of the world, formed under a generative model based on a Markov decision process, as will be described below.

Under this approach, different types of exploitative and exploratory behaviour emerge. The key aspect that motivates goal-directed uncertainty reduction is the mapping from (hidden) states to observations. This form of uncertainty reduction becomes relevant in partially observable problems, where in addition to inferring the best policy; agents also have to infer the current (hidden) state that caused an observation. In order to minimise uncertainty about the current state, agents can try to navigate to (observable) outcomes, where the mapping to the underlying hidden state is unambiguous. A simple example is a bird that is searching for prey: in the case of high uncertainty about the prey's location, a bird might go to a vantage point first to minimise uncertainty about the prey's location (i.e. the underlying hidden state), before predation. Another example is contextual inference, where an agent needs to disclose the current context (i.e. the hidden state), in order to infer what to do (e.g. is there milk in the fridge?). In case of contextual uncertainty, agents will prefer to sample outcomes that allow for precise inference about the current context (e.g. sample the fridge), before making a choice about whether to look for reward (e.g. whether or not to make tea). Formally, this means that agents will try to actively sample outcomes that have an unambiguous (low conditional entropy) mapping to hidden states – hence *active inference* allowing for 'hidden state exploration'.

Importantly, the exact same imperatives apply to beliefs about model parameters that describe a subject's knowledge about state transitions or the probability of various outcomes given the underlying (hidden) states. In other words, uncertainty about states of the world is accompanied by uncertainties about the contingencies that underwrite state transitions and the relationship between hidden states and observable outcomes. In contrast to the examples above, which reflect uncertainty about the underlying hidden state, given an agent's model of the task this form of uncertainty reflects an agent's ignorance about the causal structure of the model per se. For example, agents can be uncertain about the current context that determines the value of options (*uncertainty about a hidden state*, 'is there milk in my tea?') or uncertain about the value of options given a current context (*uncertainty about model parameters*, 'what does milky tea taste like?'). To reduce the latter type of uncertainty, agents can expose themselves to observations that complete 'knowledge gaps' and thereby learn the probabilistic structure of unknown and unexplored (novel) contingencies – hence *active learning* allowing for 'model parameter exploration'.

In the following, we introduce the theoretical framework underlying active inference and active learning and use simulations to illustrate the emergence of these particular types of exploratory behaviour. We consider the resolution of uncertainty about *states* and *parameters* in terms of *salience* and *novelty* respectively; where 'salience is to inference' as 'novelty is to learning'. We use a simple two-armed bandit problem in which a subject has to choose between a risky high reward and a safe low reward, where the probabilities of the risky option are unknown. Minimising expected free energy leads to curiosity-driven *active learning* that initially favours the novel risky option, because this option provides uncertainty reduction about an agent's parameterisation of the task. We show how the same computational framework motivates *active inference* in situations where certain actions disclose salient information about hidden states, such as whether there is currently a high or low reward probability in the risky option. Based on this paradigm, we illustrate different sorts of explorative behaviour, contrast them with random exploration or purely exploitative choices, and consider how different tendencies emerge under different priors over beliefs about outcomes and the precision of those beliefs.

## Results

### A generative model of a Markov decision process

Our theoretical approach assumes that agents, such as brains or economists, minimise the expected free energy of future outcomes and hidden states. This premise allows us to derive generic update rules for action (i.e. policy selection), perception, and learning based on variational Bayes, which is described in more detail in the Materials and methods section and previous work (*Friston et al., 2013*; *Friston et al., 2017a*). In the following, we provide a brief conceptual outline of this computational architecture to frame the discussion of active inference and active learning in the remainder of the paper.

Active inference and active learning rest upon a generative model of observed outcomes as illustrated in *Figure 1*. This generative model is used to infer the most likely causes of outcomes, that is what is the most likely true (hidden) state of the world (e.g. a current context) that caused a given observation (e.g. a win or a loss). These states are called *hidden* because they are usually not or only partially observable and can only be inferred through observations. Inferring beliefs about hidden states (i.e. state estimation) is cast as an optimisation problem based on minimising variational free energy, which finds the most likely (posterior) expectations about states of the world, given current observations. This is the same optimisation found in machine learning, where (negative) free energy is known as an evidence lower bound (ELBO). Importantly, agents can also infer different policies, defined as sequences of actions, that determine the most likely observations they will make. This means that observations depend upon policies, which requires the generative model to infer expectations about future outcomes under different policies. Thus, in addition to forming posterior beliefs about hidden states, active inference and learning rest on posterior expectations about the best policy to pursue in a given context. In other words, agents are assumed to infer 'what is the current state of the world' and 'what are the best actions to pursue' based on the same generative model of the environment.

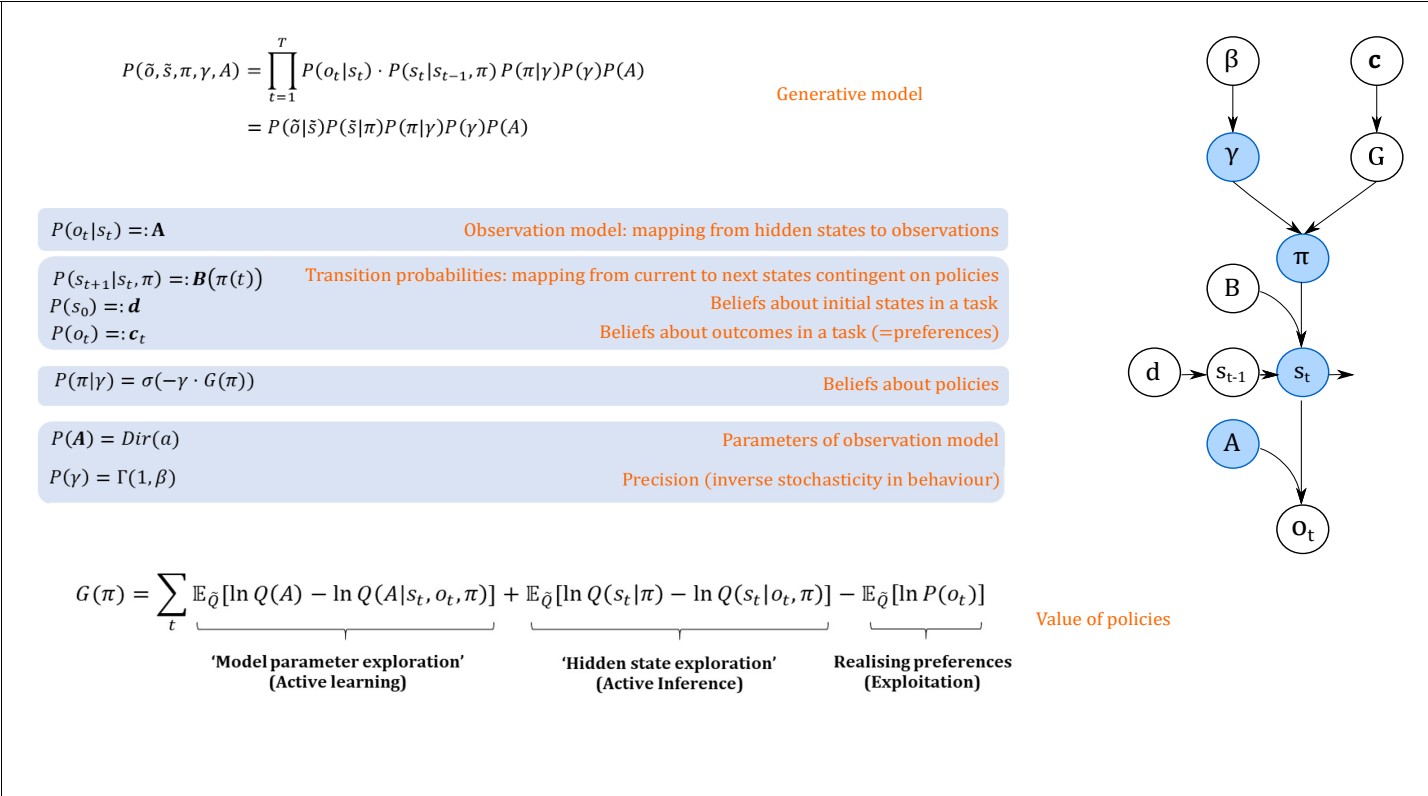

**Figure 1.** Generative model. A generative model specifies the joint probability of observations and their hidden causes. The model is expressed in terms of an *observation model* (*likelihood function*, that is the probability of observations given true states) and priors over causes. Here, this likelihood is specified by a matrix (**A**) whose rows are the probability of an outcome under all possible hidden states, $P(o_t|s_t)$. The (empirical) priors in this model pertain to transitions among hidden states (**B**) that depend upon policies (i.e. sequences of actions), $P(s_{t+1}|s_t, \pi)$ and beliefs about policies contingent on an agent's precision or inverse randomness, $P(\pi|\gamma)$, as well as (full) priors on precision (specified by a Gamma distribution) and an agent's observation model (specified by a Dirichlet distribution). The key aspect of this generative model is that policies are more probable a priori if they minimise the (sum or path integral of) expected free energy $G(\pi)$. This implies that policies become valuable if they maximise information gain by learning about model parameters (first term) or hidden states (second term) and realise an agent's preferences. Approximate inference on the hidden causes (i.e. the current state, policy, precision and observation model) proceeds using variational Bayes (see Materials and methods). Right side depicts the dependency graph of the generative model, with blue circles denoting hidden causes that can be inferred. $\sigma$=Softmax function, Dir = Dirichlet distribution, $\Gamma$= Gamma distribution, Q = (approximate) posterior. See Materials and methods for details.

DOI: https://doi.org/10.7554/eLife.41703.002

Variational free energy is a function of observations and probabilistic beliefs about hidden states (see Materials and methods), and can be understood as a statistical quantity that measures the mismatch (i.e. 'surprise') between true observations and predictions about those observations under the generative model. Minimising this mismatch ensures that these beliefs approximate the true states, given observations and maximises the (negative log) evidence of an agent's generative model. The key assumption of active inference and active learning is that we can apply the same logic to inference about hidden states *and policies*. In the context of inference about policies ('what am I going to do now?"), valuable policies are those that minimise variational free energy expected under that policy, that is the mismatch between preferred and predicted outcomes, under a given policy. Effectively, this compels agents to select policies that avoid surprising outcomes. This (expected) free energy is a proxy for surprise or model evidence, and thus allows one to cast choice behaviour as minimising expected surprise or, equivalently, maximising expected model evidence (*Friston et al., 2015*). This provides a formal grounding for the notion of the 'value' of a policy: the value is defined with respect to an agent's generative model of the world, and valuable policies maximise the expected log-evidence of that model.

Importantly, as illustrated in *Figure 1* (see Materials and methods section, *Equation 7* for details), the value (goodness) of a policy G is determined by both *extrinsic* reward and *intrinsic* information

gain. Depending on an agent's prior uncertainty about the world, gaining information can refer to exploring hidden states underlying observations, that is 'active inference', or exploring the correct parameterisation of the agent's world model, that is 'active learning'. Interestingly, these two tendencies can make opposing predictions about behaviour. Active learning allows for 'model parameter exploration' and compels agents to actively seek novel combinations of hidden states and outcomes to learn about the way in which outcomes are generated. Active inference allows for 'hidden state exploration' and compels agents to actively seek (known) salient observations that enable them to infer the underlying hidden states unambiguously. For example, if an agent is certain that a risky option has a 0.5 probability of being rewarded, this 'certain ambiguity' will be aversive (depending on her risk preferences, see appendix I). However, if she is uncertain about a 0.5 probability, however, this 'uncertain ambiguity' means there is an opportunity to resolve uncertainty and motivate active learning. In the following, we will explore this dialectic between 'active learning' and 'active inference' and speculate about their behavioural and neuronal underpinnings.

We will restrict our discussion of active inference and active learning to the context of discrete-time Markov Decision Processes (*Friston et al., 2013*; *Friston et al., 2015*). In this setting, agents are assumed to perform approximate inference based on variational Bayes, which casts a difficult and usually intractable inference problem as a bound optimisation problem (*Beal, 2003*; *Bogacz, 2017*; *Gershman, 2019*). This implies that expectations about hidden states are updated to minimise variational free energy under a generative model. *Figure 1* provides an illustration of the Markovian generative model used in the simulations below. Observable outcomes $o_t$ at a particular discrete time-step depend upon true hidden states $s_t$ in the world, while hidden states evolve according to Markovian transition probabilities contingent upon actions emitted by an agent. The generative model is specified by two sets of arrays. The first, $A$, maps from hidden states to outcomes. That means that $A$ models an agent's observation model or the emission function in a hidden Markov model, specifying the likelihood of an observation under a given hidden state. The second, $B$, prescribe the transitions among hidden states, contingent on a policy $\pi$. These transitions are Markovian, such that the probability of the subsequent state is fully determined by the current state and action. The arrays **c** and **d** encode prior expectations about observations, and initial states, respectively. The former specifies an agent's preferences or utilities over outcomes and determines the 'extrinsic' reward component of a policy's value, whereas the latter specifies an agent's prior beliefs about the starting point in a task. We refer to the Appendix I for a more detailed discussion of the role of **c** and **d** in exploitative and exploratory behaviour. Finally, the precision $\gamma$ reflects an agent's stochasticity or randomness in behaviour. This precision term is parameterised by a rate parameter $\beta$, such that the expected value of $\gamma$ is $\frac{1}{\beta}$. Note that under this generative model, $\gamma$ is a hidden state that can be inferred. In the following simulations, however, we will focus on the role of $\beta$ in determining an agent's overall level of stochasticity (i.e. '*random exploration*') in behaviour, but we discuss time-dependent updates of precision in the section on potential neuronal correlates of active inference and active learning (section 'Behavioural and neural predictions').

In the following simulations of active learning and active inference (available online, cf. *Schwartenbeck, 2019a*), we focus on the two kinds of information gain, namely, foraging for information about the correct parameterisation of the observation model (active learning or 'model parameter exploration') and using the observation model to accurately infer hidden states (active inference or 'hidden state exploration'). We will assume that state transitions as well as the number and type of observations and initial states are already learned. How the state space and the dimensions of the different matrices that determine the mapping between these states are themselves learned is an important and interesting question but goes beyond the scope of this paper (see *Laversanne-Finot et al., 2018* for a discussion of curiosity-driven learning of goal states, for instance).

## Model parameter exploration via active learning

In this section, we simulate the effects of active learning or 'model parameter exploration' on behaviour (first term in *Equation 7*, Materials and methods section and *Figure 1*). The aim of this section is to characterise the behavioural phenotype of active learning in different task settings, and contrast this type of goal-directed exploration with random exploration. We simulate a simple experiment, where an agent has to choose between a safe and a risky option, such as a rat in a T-shaped maze seeking reward in one of two goal arms. . We assume that the agent knows that it can only sample

one of the two arms and that one arm (left arm in *Figure 2*) contains a certain small reward whereas the other arm (right arm in *Figure 2*) contains an uncertain, high reward. Importantly, however, the agent does not know about the reward probabilities in the uncertain arm in the beginning of the experiment, but can learn about these contingencies by updating its observation model via experience-dependent learning. Learning the observation model (i.e. building the A-matrix) is cast as updating the concentration parameters of a Dirichlet distribution that specifies the mapping from hidden states to observations (see Materials and methods for details). These updates effectively reflect normalised counts of experienced particular state-outcome mappings, as will be illustrated below.

**Figure 2.** Generative Model of a T-maze task, in which an agent (e.g. a rat) has to choose between a safe option (left arm) and an ambiguous risky option (right arm). There are three different states in this task reflecting the rat's location in the maze; namely, being located at the starting position or sampling the safe or risky arm. Further, there are four possible observations, namely being located at the starting position, obtaining a small reward in the safe option, obtaining a high reward in the risky option and obtaining no reward in the risky option. (A) The A-matrix (*observation* or *emission* model) maps from hidden states (columns) to observable outcome states (rows, resulting in a 4x3 matrix). There is a deterministic mapping when the agent is in the starting position or samples the safe reward. When the agent samples the risky option, there is a probabilistic mapping to receiving a high reward or no reward. The A-matrix depends on concentration parameters $a$ that are updated due to observing transitions between states and observations (in this example: receiving a high or no reward in the risky option), where $a_0$ reflects the prior concentration parameters without having made any observation yet (prior to normalisation over columns). (B) The B-matrix encodes the transition probabilities, that is the mapping from the current hidden state (columns) to the next hidden state (rows) contingent on the action taken by the agent. Thus, one needs as many B-matrices as there are different policies available to the agent (shown here: choose safe or choose risky). Here, the action simply changes the location of the agent. (C) The c-vector specifies the preferences over outcome states. In this example, the agent prefers (expects) to end up in a reward state and dislikes to end up in a no reward state, whereas it is somewhat indifferent about the 'intermediate' states. Note that these preferences are represented as log-probabilities (to which a softmax function is applied). For example, these preferences imply that visiting the high reward state is $\exp(4) \approx 55$ times more likely than the starting point ($\exp(0) = 1$) at the end of a trial. The d-vector specifies beliefs about the initial state of a trial. Here, the agent knows that its initial state is the starting point of the maze.

DOI: https://doi.org/10.7554/eLife.41703.003

## Model structure

To simulate behaviour, one needs to specify the parameterisation of the generative model, which has been described in detail in previous work (*Friston et al., 2016*). In this task, we need to define a hyperprior on the precision of policy (choice) selection ($\beta$ in *Figure 1*), which reflects the randomness in policy selection. Unless otherwise specified, we have set $\beta$ to a (standard rate parameter) value of 1. As shown in *Figure 2*, we define three different states in this task – as determined by the rat's location in the maze; namely, being located at the starting position or sampling the safe or risky arm. Further, we define four possible observations; namely, being located at the starting position, obtaining a small reward in the safe option, obtaining a high reward in the risky option and obtaining no reward in the risky option. The ***A***-matrix (observation model) then determines the mapping from states to observations, while the ***B***-matrix (transition probabilities) specifies the mapping between hidden states given an action (which we assume to be learned). Further, we need to specify an agent's expectations over observations that reflect its preferences. These expectations are encoded in a **c**-vector, which we have set to $c = \begin{bmatrix} 0 & 2 & 4 & -2 \end{bmatrix}$ in the following simulations, reflecting an agent's preference for being in the starting position, obtaining a safe reward, obtaining a high reward and obtaining no reward in a risky option, respectively. Note that here and below these preferences are defined as an agent's *log-expectations* over outcomes, which means that preferences are passed through a softmax function and correspond to log probabilities (giving $c = \begin{bmatrix} -4.15 & -2.15 & -0.15 & -6.15 \end{bmatrix}$). For example, the definition of these preferences implies that the agent believes that visiting the high reward state is $\exp(4) \approx 55$ times more likely than visiting the starting point ($\exp(0) = 1$) at the end of a trial. The **d**-vector encodes an agent's expectations about the initial state, which was defined to reflect full certainty about starting each trial in the starting position of the maze. In simulations that include learning, we set the initial concentration parameters for obtaining a high reward (or not) to 0.25 (i.e. position (3,3) and (4,3) in the A-matrix in *Figure 2*), and these concentration parameters are updated according to a learning rate $\eta$, which was set to 0.5. *Figure 2* illustrates the architecture of the generative model of this task.

## Active learning

*Figure 3A* illustrates an experiment that was simulated under active learning with an underlying high-reward probability of 50% in the risky option. The bottom panel illustrates the evolution of beliefs (concentration parameters) about the underlying emission probabilities of the task for every trial of the experiment, which in turn determine policy selection as illustrated in the first panel. Note that at the start of the experiment, the agent assigns equal probability to receiving a high reward and no reward at the risky option, but these beliefs have very low certainty (i.e. very small concentration parameters). This leads the agent to explore and gather information in the beginning of the experiment by choosing the risky option, that is to *learn actively*. After trial 10, the agent (correctly) assigns a probability of 50% to a high reward in the risky option, but now with higher confidence (i.e. larger concentration parameters). Consequently, the agent now prefers to exploit and sample the safe option, driven by both the expected value of this option and a preference for visiting unambiguous states. Note that this result also depends on the agent's risk preferences, as discussed in Appendix I.

*Figure 3B* illustrates the same task but with a reward probability of 0.75 for the risky option. Here, after a similar number of exploration trials as in *Figure 3A*, the agent becomes confident that it should select the risky option, given its higher expected value. This can be seen by the fact that the agent continues to select the risky option (blue dots) with high confidence (shaded area behind blue dots, first panel) because the risky option is mostly rewarded (green dots, second panel).

## The role of stochasticity in active learning

The above simulations highlight an important aspect of exploratory behaviour, namely behaviour that is goal-directed and aims at reducing uncertainty about a specific part of an agent's model, in this example the part of the A-matrix (i.e. the observation or emission function) that specifies the mapping from sampling the risky option to obtaining a high or low reward. This means that the agent tries to gain insight into a particular part of the structure of world that it is unsure about. Importantly, this predicts that this sort of exploratory behaviour will be most prevalent if there is high uncertainty about the structure of a task, such as in the beginning of a game. This also suggests

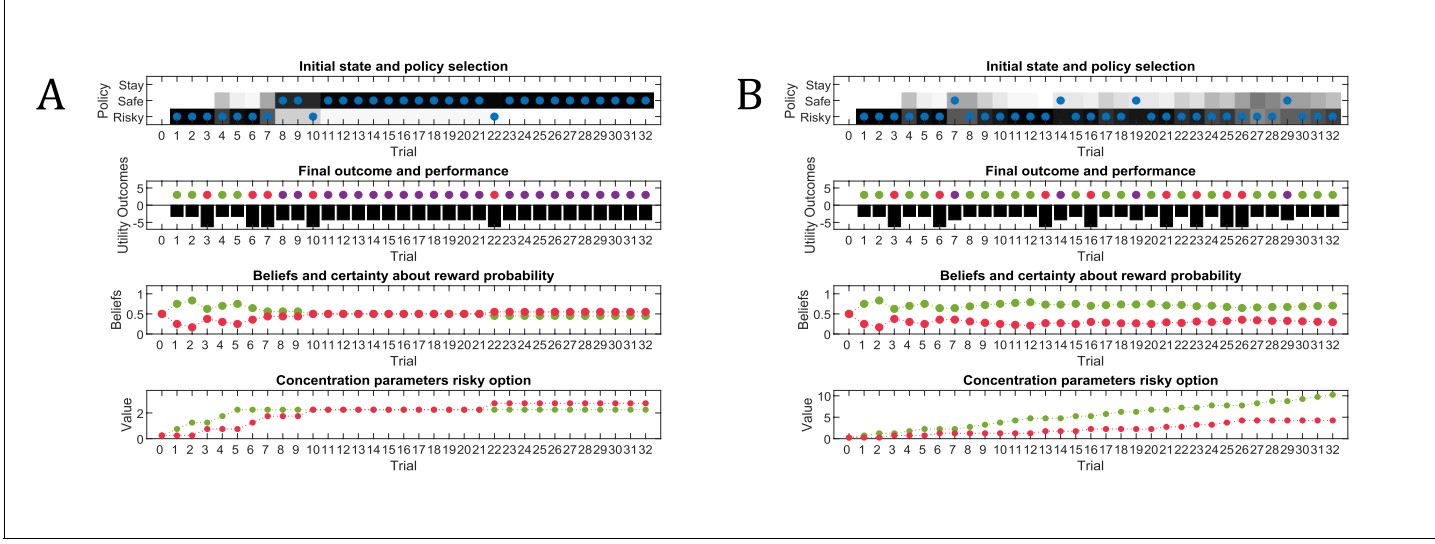

**Figure 3.** Simulated responses during active learning. This figure illustrates responses and belief updates during a simulated experiment with 32 trials. The first panel illustrates whether the agent sampled the safe or risky option as indicated by the blue dots, as well as the agent's beliefs about which action to select. Darker background implies higher certainty about selecting a particular action. The second panel illustrates the outcomes at each trial and the utility of each outcome. Outcomes are represented as coloured dots, where purple refers to a small and safe reward, green to a high reward and red to no reward in the risky option. Black bars reflect the utilities of the outcome. Note that these utilities are defined as log-probabilities over outcomes (see main text and *Figure 2*), thus a value closer to zero reflects higher utility of an outcome. The third panel illustrates the evolution of beliefs about the reward probabilities in the risky option (red = belief about no reward, green = belief about high reward). The fourth panel illustrates the evolution of the corresponding concentration parameters of the observation model over time (red = concentration parameter for the mapping from risky option to no reward, green = concentration parameter for the mapping from risky option to high reward, cf. *Figure 2A*). (A) In this example, the simulated agent makes predominantly curious and novelty-seeking choices in the beginning of the experiment. After the tenth trial, the agent is confident that the risky option provides a probability of 0.5 for receiving a high reward, which compels it to choose the safe option afterwards. (B) Same setup as in (A), but now the true reward probability of the risky option is set to 0.75. After sampling the risky option in the beginning of the experiment and learning about the high reward probability of that option, the agent becomes increasingly certain that the risky option has a high probability of a reward. This compels the agent to continue sampling the risky option and only rarely visiting the safe option with low certainty, as illustrated in panel one.

DOI: https://doi.org/10.7554/eLife.41703.004

an important confound when investigating the influence of reward and uncertainty on behaviour; namely, the fact that the rewarding options will often be associated with the lowest uncertainty because they are sampled most frequently (*Wilson et al., 2014a*). This confound highlights the importance of analysing behaviour at the beginning of an experiment when there is high uncertainty about all available options (*Gershman, 2018b*; *Gershman, 2018a*).

As illustrated earlier, goal-directed information-gain can be contrasted with random exploration, such as in $\epsilon$-greedy or softmax choice rules where the degree of randomness is governed by an inverse temperature parameter (*Sutton and Barto, 1998a*). In its simplest form, random exploration implies that exploratory behaviour will not be informed by an agent's uncertainty about different options or its uncertainty about different parts of the world. This implies that such behaviour will not decrease uncertainty per se but may cause 'accidental' belief-updating due to random or stochastic selection of different policies. Here, this sort of behaviour is controlled by the precision of policy selection (see *Equation 4* in Materials and methods). This means that random exploration can be understood as imprecise behaviour. Importantly, the precision of behaviour does not depend on an agent's uncertainty about the world, such that there is no predicted relationship between 'random exploration' and the time-course of an experiment (see below). *Figure 4* illustrates the effects of highly imprecise ($\beta = 2^3$, *Figure 4A*) and highly precise ($\beta = 2^{-3}$, *Figure 4B*) types of behaviour. Note that the expected value of precision is the inverse of $\beta$, that is $E(\gamma) = \frac{1}{\beta}$ (*Figure 1*).

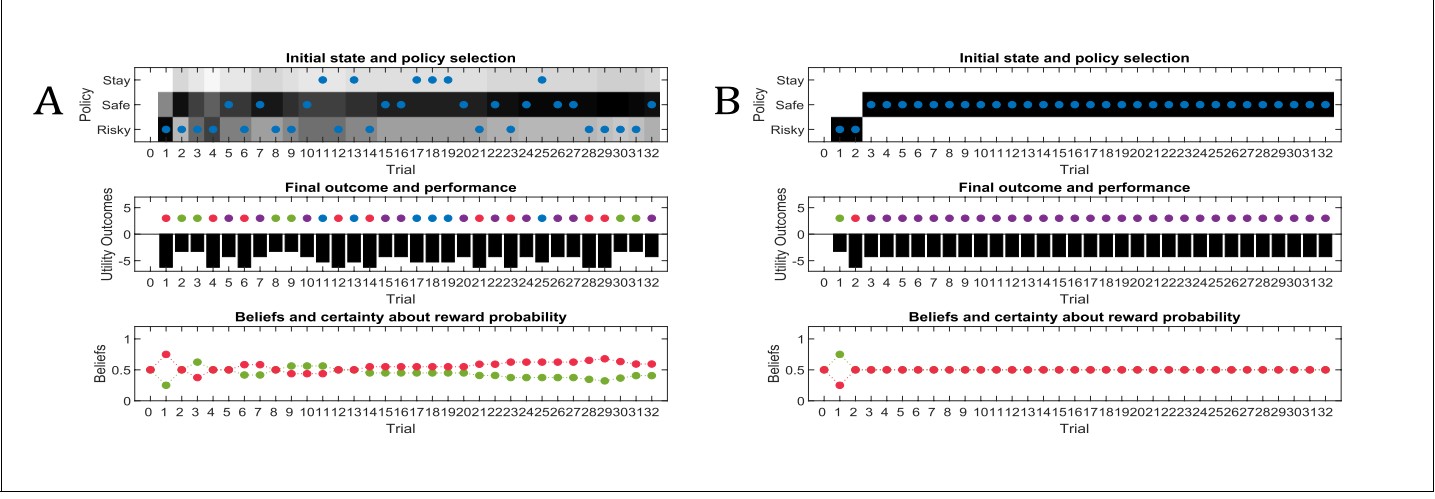

**Figure 4.** Effects of precision on behaviour. Same setup as in *Figure 3*, but now with varying levels of stochasticity. (**A**) A high degree of *random* exploration results from very imprecise behaviour ($\beta = 2^3$), whereas (**B**) highly precise behaviour ($\beta = 2^{-3}$) results in very low randomness in behaviour.

DOI: https://doi.org/10.7554/eLife.41703.005

## Broken 'active learning'

Active learning predicts that the ability to learn about the environment and minimise uncertainty is a determining factor of the value of policies. This can be illustrated by disabling the influence of active learning on policy evaluation, as shown in *Figure 5*. In this case, policies cannot be distinguished in terms of their uncertainty reduction about model parameters. Consequently, the value of policies is determined by visiting preferred and unambiguous outcomes. This means that agents will not exhibit active learning, and the only way to learn about the environment is by accidently (randomly) sampling a non-preferred option. *Figure 5* illustrates this problem: here, the true reward probability of

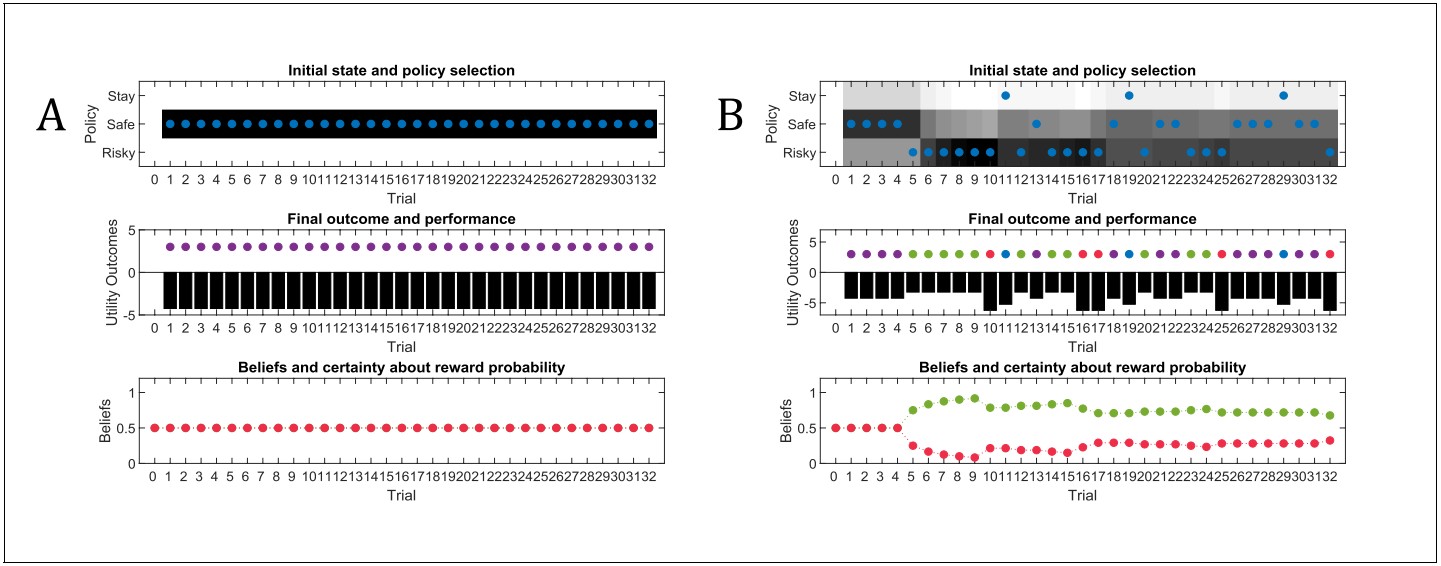

**Figure 5.** 'Broken' active learning (parameter exploration). Same setup as in *Figure 3*, but now with a true reward probability of 0.75 and no active learning as a determinant of the value of policies (first term of *Equation 7*, Materials and methods section). (**A**) If behaviour is very precise ($\beta = 2^{-3}$), the agent will never find out that the risky option is more preferable than the safe option, because there is no active sampling of its environment. (**B**) In contrast, if the agent's behaviour has a higher degree of randomness (low precision, $\beta = 2^3$), then it will eventually learn about the reward statistics in the risky option from randomly sampling this alternative, and infer that it is preferable over the safe option.

DOI: https://doi.org/10.7554/eLife.41703.006

the risky option is 0.75, but in the absence of any active learning, the agent can only find out about the value of the risky option by randomly sampling this alternative. Thus, if the agent shows very precise (non-random) behaviour (*Figure 5A*), it is very unlikely to discover that the risky option is better than the safe option, and only by showing very imprecise behaviour (*Figure 5B*) the agent will be able to develop a (weak) preference for the risky option. This illustrates an intriguing point, namely that random exploration may serve an adaptive function in the absence of goal-directed exploratory behaviour, for example due to an agent's inability to evaluate its uncertainty about the world.

## Time courses of exploratory behaviour

A general problem when investigating the role of exploration in value-based decision-making is that if an agent is allowed to move around freely, there will be a relationship between the reward statistics of an option and its associated uncertainty. Rewarding arms will be associated with a lower level of uncertainty simply because they are sampled more often (*Gershman, 2018a*; *Gershman, 2018b*; *Wilson et al., 2014a*). To compare different computational architectures that might underlie exploratory behaviour and information-gain, it is therefore important to investigate the time-course of behaviour, as illustrated in *Figure 6* based on a true reward probability of 0.5 in many simulations of the active learning task. *Figure 6A* illustrates the time-course of behaviour under active learning conditioned on the concentration parameters of the A-matrix (observation model,)

Unsurprisingly, the agent strongly prefers to choose the risky option when she believes that the reward probability is high (right bottom corner in *Figure 6A*) and strongly prefers to choose the safe option if the probability of a high reward is low (left upper corner in *Figure 6A*). Importantly, we also observe a gradient across the diagonal, such that agents have a strong preference to choose the risky option if there is high uncertainty about its reward contingencies (i.e. both concentration parameters of the **A**-matrix are low, lower left corner in *Figure 6A*). In contrast, the probability to choose the risky option is very low if the agent is very certain that the probability to receive a high reward is 0.5 (i.e. both concentration parameters of the **A**-matrix are high, upper right corner in *Figure 6A*). In line with this, the probability of choosing the risky option over time under active learning shows that there is a very high preference for sampling the risky (uncertain) option in the beginning of a trial, which then monotonically decreases over time (*Figure 6C*).

*Figure 6B* illustrates the time-course of behaviour without active learning but with a high degree of random exploration (low prior precision), where the only way to learn about the true reward probabilities is by randomly sampling the risky option. The pattern of *Figure 6B* reflects a noisier version of *Figure 6A*. Aside from the larger randomness in behaviour, there is also an important difference when uncertainty about the true reward statistics is high (lower left corner): in the absence of active learning, there is no preference for the risky option when the relevant concentration parameters of the A-matrix are both low (lower left corner of *Figure 6B*). This also becomes apparent when looking at the time course of choosing the risky option, such that there is no initial preference for the risky option reflecting uncertainty reduction in the beginning of a trial (*Figure 6C*). Rather, the probability to select the risky option remains relatively stable across trials and reflects the overall level of randomness in behaviour. If there is no learning at all (i.e. the concentration parameters of the A-matrix do not change), the probability to choose the risky option is constant and simply reflects the stochasticity of individual behaviour (*Figure 6C*).

## Hidden state exploration via active inference

In this section, we illustrate a second type of behaviour that aims at gaining information about the world, namely exploring about hidden states of a task, as reflected by the second term of *Equation 7* (Materials and methods section). In contrast to 'model parameter exploration', which motivates *active learning* to reduce uncertainty about an agent's model of the world, 'hidden state exploration' motivates *active inference* to form accurate beliefs about the current state of the world, based on an agent's model of the task. One example of this behaviour is inferring the current context, which we illustrate in the following simulations, using a slightly adjusted version of the previous task. We now assume that the agent has learned that she could be in two possible (hidden) states in this task, namely either in a context where the risky option provides high or low probability for obtaining a reward, but this contextual information is hidden from her. However, in this version of the task, she

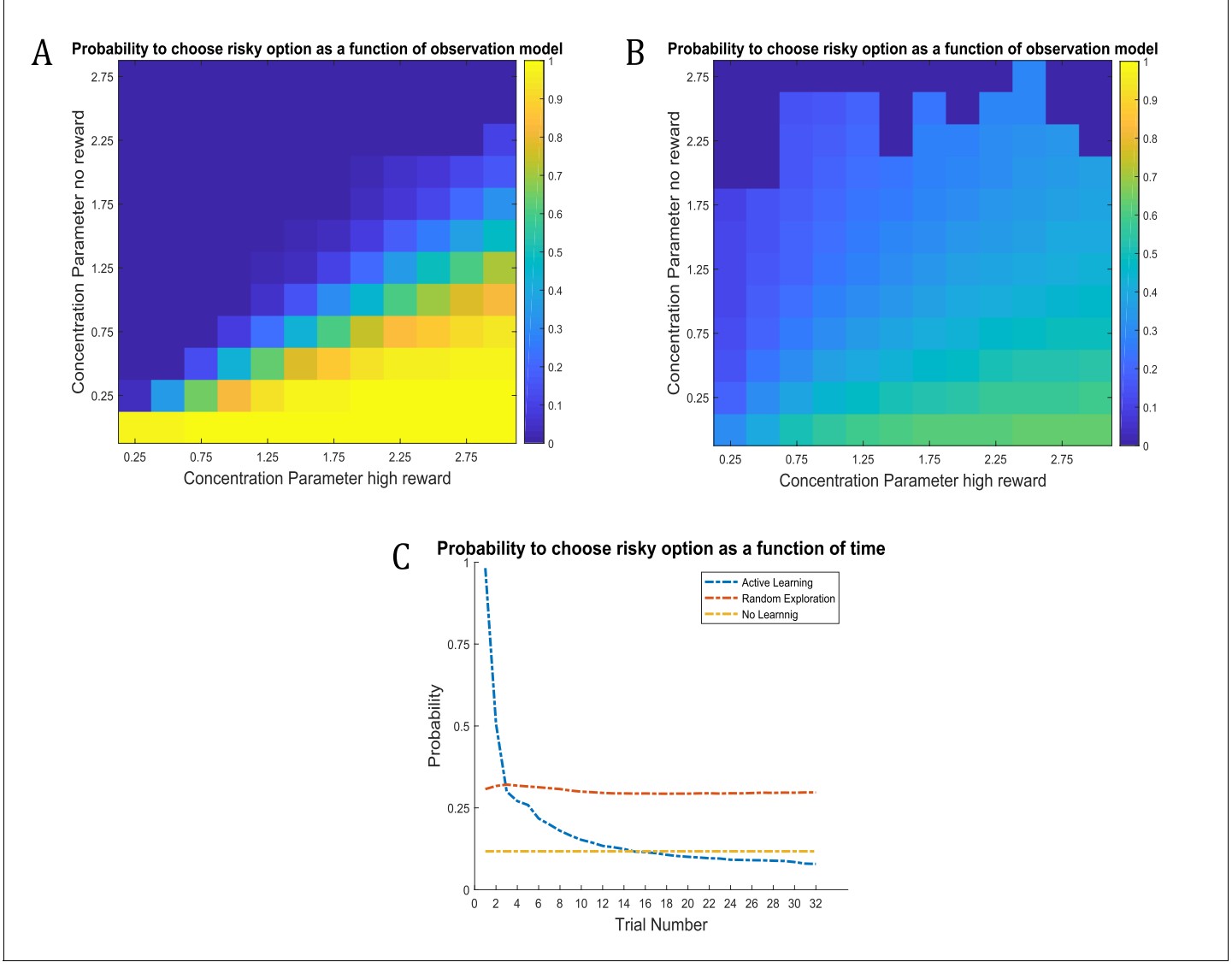

**Figure 6.** Time-course of active learning and random exploration. Simulations of 1000 experiments with 32 trials each under a true reward probability of 0.5. (A) Probability to choose the risky option as a function of the concentration parameters for high reward and no reward in the risky option under active learning. The probability to choose the risky (uncertain) option is high if there is high uncertainty about this option at the beginning of a task. Note how the probability of choosing the risky option decreases as the agent becomes more certain that the true reward probability of the risky option is 0.5 (gradient along the diagonal). (B) When there is no active learning but high randomness (low prior precision, $\beta = 2^3$), there is no uncertainty-bonus for the risky option if the agent is uncertain about the reward mapping (lower left corner). The probability to sample the risky option increases only gradually with increasing certainty about a high reward probability (gradient along x-axis). (C) Average probability to choose the risky option as a function of time for active learning (as in A), random exploration (as in B) and in the absence of any learning. Active learning induces a clear preference for sampling the informative (risky) option at early trials. In contrast, random exploration without active learning does not induce a preference for uncertainty-reduction at early trials, and the probability to choose the risky option quickly converges as the estimate of the true reward probability converges to 0.5 due to random sampling of the risky option. In the absence of any learning, the probability to choose the risky option is constant and reflects the precision or randomness in an agent's generative model (simulated with, $\beta = 2^1$).

DOI: https://doi.org/10.7554/eLife.41703.007

can also choose to sample a cue before choosing the safe or risky option, which tells her about the reward probabilities (i.e. context) of the current trial.

**Figure 7.** Generative model of a T-maze task, in which an agent (e.g. a rat) has to choose between a safe option (left arm) and a risky option (right arm). In contrast to the previous task, the rat can now be in two different contexts that define the reward probability of the risky option, which can be high (75%) or low (25%). Besides sampling the safe or risky option, it can now also sample a cue that signifies the current context. This results in a state space of eight possible states, defined by the factors location (starting point, cue location, safe option, risky option) and context (high or low reward probability in risky option). Further, there are seven possible observations the agent could make, namely being at the starting position, sampling the safe option, obtaining a/no reward in the risky option, and sampling the cue that indicates a high/low reward probability. (**A**) The A-matrix (*observation* or *emission* model) maps from hidden states (columns) to observable outcome states (rows, resulting in an 8 × 7 matrix). There is a deterministic mapping when the agent is in the starting position, samples the safe reward or samples the cue. When the agent samples the risky option, there is a probabilistic mapping to receiving a high reward or no reward that depends on the current context. In contrast to the previous example, no updates of the A-matrix take place in this task. (**B**) The B-matrix encodes the transition probabilities, that is the mapping from the current hidden state (columns) to the next hidden state (rows) contingent on the action taken by the agent, which simply changes the location of the agent. For simplicity, only the transition probabilities for the factor location are shown, which replicate across the two contexts (resulting in an 8 × 8 transition matrix). (**C**) The c-vector specifies the preferences over outcome states. In this example, the agent prefers (expects) to end up in a reward state and dislikes to end up in a no reward state, whereas it is indifferent about the 'intermediate' states (starting position or cue location). The d-vector specifies beliefs about the initial state of a trial. Here, the agent knows that its initial state is the starting point of the maze, but has a uniform prior over the two contexts. In experiments where the context is stable, this uniform prior can be updated to reflect experience-dependent expectations about the current context.
DOI: https://doi.org/10.7554/eLife.41703.008

## Model structure

The generative model of the 'hidden state exploration' task is illustrated in *Figure 7*. We have used the same formalisation as in the previous model, except that the agent now performs inference about sampling the safe or risky option directly, or sampling a cue first that signifies the current context, namely a high (75%) or a low (25%) probability to obtain a reward in the risky option. In comparison to the previous generative model illustrated in *Figure 2*, this increases the size of the state space by the additional cue location and the (hidden) context factor, resulting in eight different

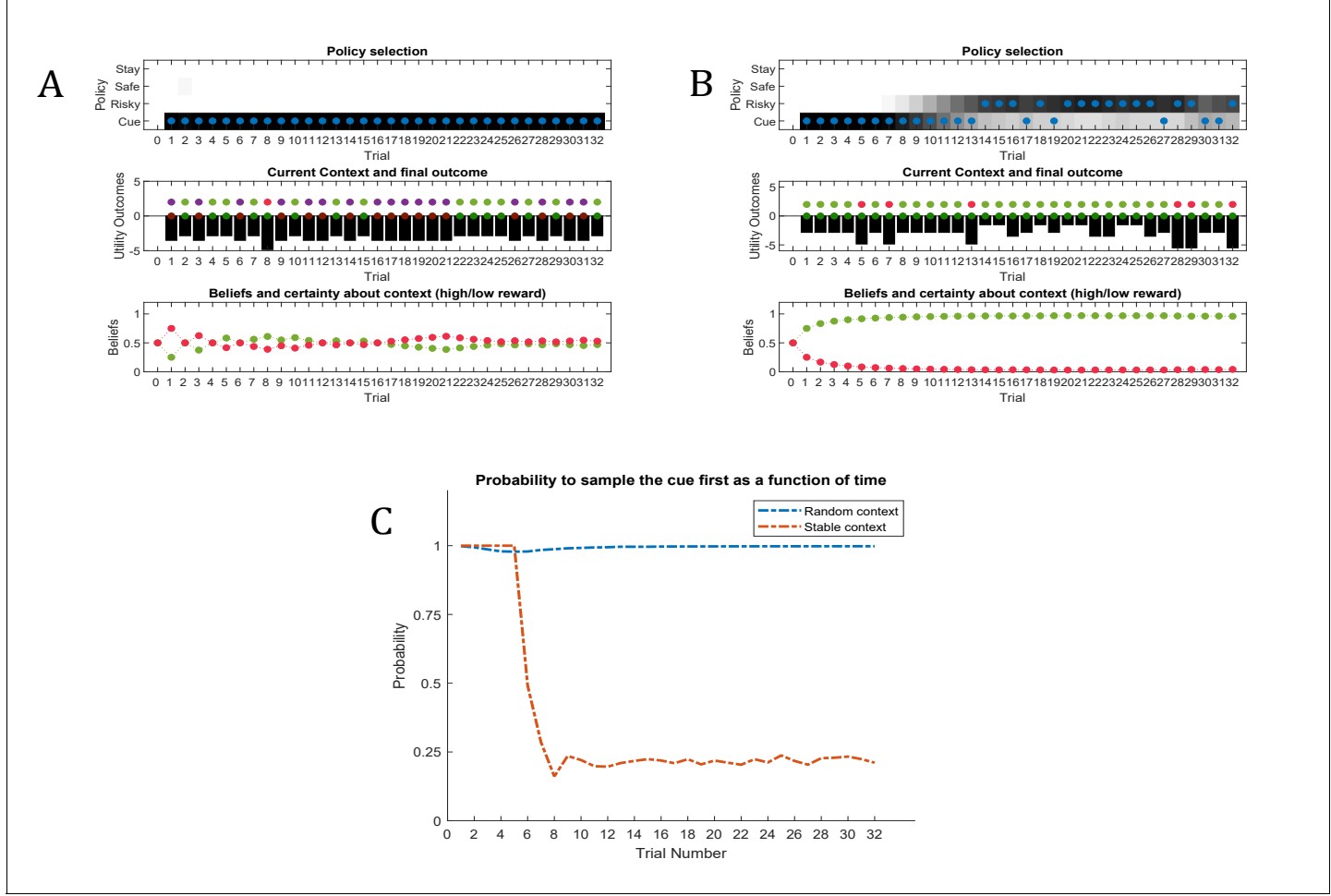

**Figure 8.** Simulated responses during inference. In this experiment, the current context indicates either a high (75%) or low (25%) reward probability in the risky option. The agent can gain information about the current context of a trial by sampling a cue, which signifies the current context. (A) Simulated experiment with 32 trials and a random context that changes on a trial-by-trial basis: the first panel illustrates the choice of the agent at the beginning of a trial and the agent's beliefs about action selection (darker means more likely). Note that the agent always chooses to sample the cue first before choosing the safe or risky option. The second panel illustrates the outcomes of every trial (purple = safe option, green = high reward in risky option, red = no reward in risky option) and their utilities (black bars, closer to zero indicates higher utility). Note that a green or red outcome indicates that the agent has chosen the risky option after sampling the cue. Dark red and green dots indicate the current context as signified by the cue (dark red = low reward probability in risky option, dark green = high reward probability in risky option). Note that the agent only samples the risky option if the cue indicates a high reward context. The third panel shows the evolution of beliefs concerning the current state (i.e. high or low reward context). (B) Same setup as before, but now with a constant context that indicates a high reward probability in the risky option. Here, the agent becomes increasingly confident that it is in a high reward context, which compels it to sample the risky option directly after about one third of the experiment, whilst gathering information in the cue location in the first third of the experiment. (C) Time-course of the probability to sample the cue first as a function of trial number in an experiment (in 1000 simulated experiments). If the context is random, there is a nearly 100% probability to sample the cue first at every trial. In a stable context, the probability to sample the cue shows a sharp decrease once the agent has gathered enough information about the current (hidden) state.

DOI: https://doi.org/10.7554/eLife.41703.009

(hidden) states (columns of A-matrix in *Figure 7*). The B-matrix encodes the transitions between different locations from the starting position of the maze; namely, sampling the cue, the safe option, or the risky option. The c- and d-vectors are defined analogously to the previous example, except that the d-vector now reflects a uniform prior about starting the maze in one of the two contexts. We did not include any curiosity-driven learning in these simulations, except that we allowed for experience-based updates of the d-vector in one simulation (*Figure 8B*), which describe a task in which the true state of the task can be learned gradually. Updates of the (concentration parameters of the) d-vector are implemented analogously to the updates of the A-matrix in the 'parameter exploration' example above. Note that, in principle, such updates would also allow the agent to continuously learn about the current reward probabilities of the risky option without sampling the cue first, analogously to the 'model parameter exploration' example. Importantly, however, parameter exploration will not work if the context changes rapidly, such as on a trial-by-trial basis. This provides an important illustration of the different time-courses of inference and learning (see 'comparing model parameter and hidden state exploration' section below). In the following, we will illustrate active inference in a task with a volatile and a stable context, and show how an agent fails to perform goal-directed exploration of hidden states if active inference is compromised.

## Active inference

*Figure 8A* illustrates 'hidden state exploration' in an experiment, where the current context cannot be learned, that is changes randomly on a trial by trial basis. *Active inference* predicts that the agent will always sample the cue at the beginning of every trial to reduce ambiguity about the current hidden state (context) (first and third panel of *Figure 8A* and blue line in *Figure 8C*). The subsequent behaviour in a trial depends on the information obtained at the cue. If the cue signifies a context with high reward probability (dark green dots in second panel of *Figure 8A*), the agent will choose the risky option. In contrast, if the cue indicates a context with a small reward probability, she will choose the safe option.

This simulation illustrates an important difference to the active learning simulations above: in these simulations, there is nothing to be learned about the state of the world, because the current state changes randomly on a trial by trial basis. Thus, this task could not be solved by learning the reward-mapping of the risky option, because there is no knowledge about the reward statistics that could be carried over from one trial to the next. This highlights the necessity to perform trial-by-trial inference about the current state of the world, as opposed to continuous parameter learning.

*Figure 8B* illustrates simulations of the same task, but now with a stable context of a high reward probability in the risky option, allowing for experience-dependent updates of the agent's prior over initial contexts (parameters in the d-vector, cf. *Figure 7*) based on information obtained from the cue. In the first third of the experiment, we observe the same choice bias as in *Figure 8A*, namely a preference to sample the cue first before choosing the safe or risky option. In this experiment, however, the agent always obtains the same information from the cue location, indicating a stable environment with a high reward probability in the risky option. Once the agent becomes confident enough in its beliefs about the current context, it starts to sample the risky option without sampling the cue first (cf. red line in *Figure 8C*, see Appendix I for a more detailed discussion of the 'cost' of sampling the cue). Note that in contrast to the 'parameter exploration' simulations above, the agent updates its beliefs based on the (hidden state) information provided by the cue, not the actual outcome (i.e. obtaining a reward). This can be seen in the belief-updating after trial five, for instance: the agent samples the cue, which indicates a high reward probability context, and obtains no reward from the risky option. Despite the negative outcome, it increases its belief about being in the high reward context (third panel of *Figure 8B*), due to the information obtained from the cue.

## Broken 'active inference'

What happens if an agent fails to perform 'hidden state exploration'? *Figure 9* shows simulations of behaviour when information-gain about the hidden state is not considered during policy selection. This implies that the cue location has no informative value, and is equally preferable to the starting location of the maze (because they have the same utility, cf. c-vector of *Figure 8*). This results in a constant preference for the safe option, because this agent is insensitive to the informative value of the cue. In the example illustrated in *Figure 9*, the agent fails to acknowledge that there is a high

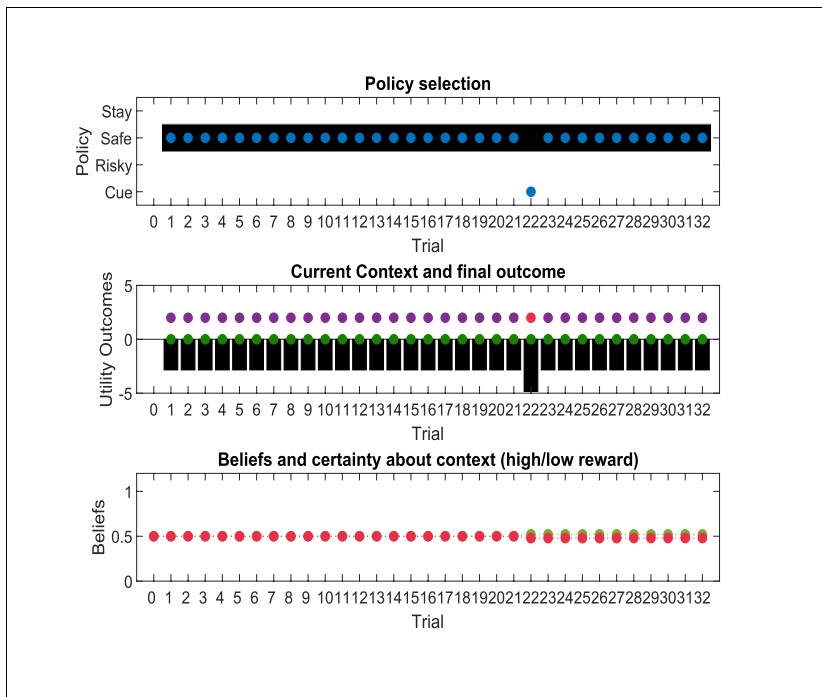

**Figure 9.** 'Broken' active inference. Same setup as in *Figure 8*, but now without a 'hidden state exploration' bias in policy selection (second term of *Equation 7*, Materials and methods section). The agent fails to learn that there is a constant high reward probability for the risky option because it does not gain information about the current hidden state (context). Consequently, it continues to prefer the safe option. The probabilities to sample different options (first panel) now simply reflect the agent's prior preferences as encoded in the c-vector (cf., *Figure 7*).
DOI: https://doi.org/10.7554/eLife.41703.010

reward probability in the risky option and continues to prefer the safe option. Analogously to *Figure 6*, the only way to sample the cue (and other options) more frequently would be by increasing the randomness in behaviour.

## Comparing model parameter and hidden state exploration

We have shown that distinct response profiles for exploratory behaviour arise from different types of uncertainties, namely uncertainty about model parameters and uncertainty about hidden states. Active learning arises when agents choose options that decrease their uncertainty about the correct parameterisation of the world, such as the reward probability in a risky option. Active inference, on the other hand, aims at gathering information about the current (hidden) state of the world, for example the current context. These behavioural tendencies can align or result in opposing predictions for behaviour in different tasks. In this section, we provide direct comparisons of *active learning* (parameter exploration) and *active inference* (hidden state exploration) in different variants of the tasks introduced above. In these simulations, we use an identical parameterisation for these two types of behaviour except that active learning is only governed by the first and third terms of *Equation 7* (model updating and realising preferences) and active inference is only governed by the last two terms of *Equation 7* (realising preferences and minimising ambiguity). We contrast these types of goal-directed exploration with a 'random exploration' agent with a higher degree of stochasticity in its behaviour (see Materials and methods for details), but no bias for (goal-directed) parameter or hidden state exploration, which serves as a baseline for the other two types of exploratory behaviour. Thus, this agent will be solely governed by the (third) realising preferences term in *Equation 7*, but can still update its model of the task due to randomly sampling different options. We compare these agents in situations where the risky option is either advantageous (reward probability of 85%) or disadvantageous (reward probability of 15%). We use the average cumulative reward in 100 simulated experiments with 32 trials each as a measure of performance for these three agents, where we

define a low reward as one food pellet and a high reward as four food pellets that could be obtained by the agent.

*Figure 10* depicts the behaviour of these three agents in the task illustrated at the top of *Figure 2*, where a rat has to choose between a certain safe and an uncertain risky option. In line with the previous simulations, we observe that the 'parameter exploration' agent quickly learns to prefer the risky option if there is a high reward probability (left upper panel of *Figure 10*) and to avoid the risky option if there is a low reward probability (right upper panel). The 'random exploration' agent also converges on these estimates, but much slower. Interestingly, we observe that the 'hidden state exploration' agent fails to adjust to the reward statistics of this task. This is because, from the perspective of this agent, there is no hidden state to explore that could be informative about the current reward statistics. The only way to learn about the statistics of the task would be by sampling observations that are a priori associated with high ambiguity. Such observations, however, are

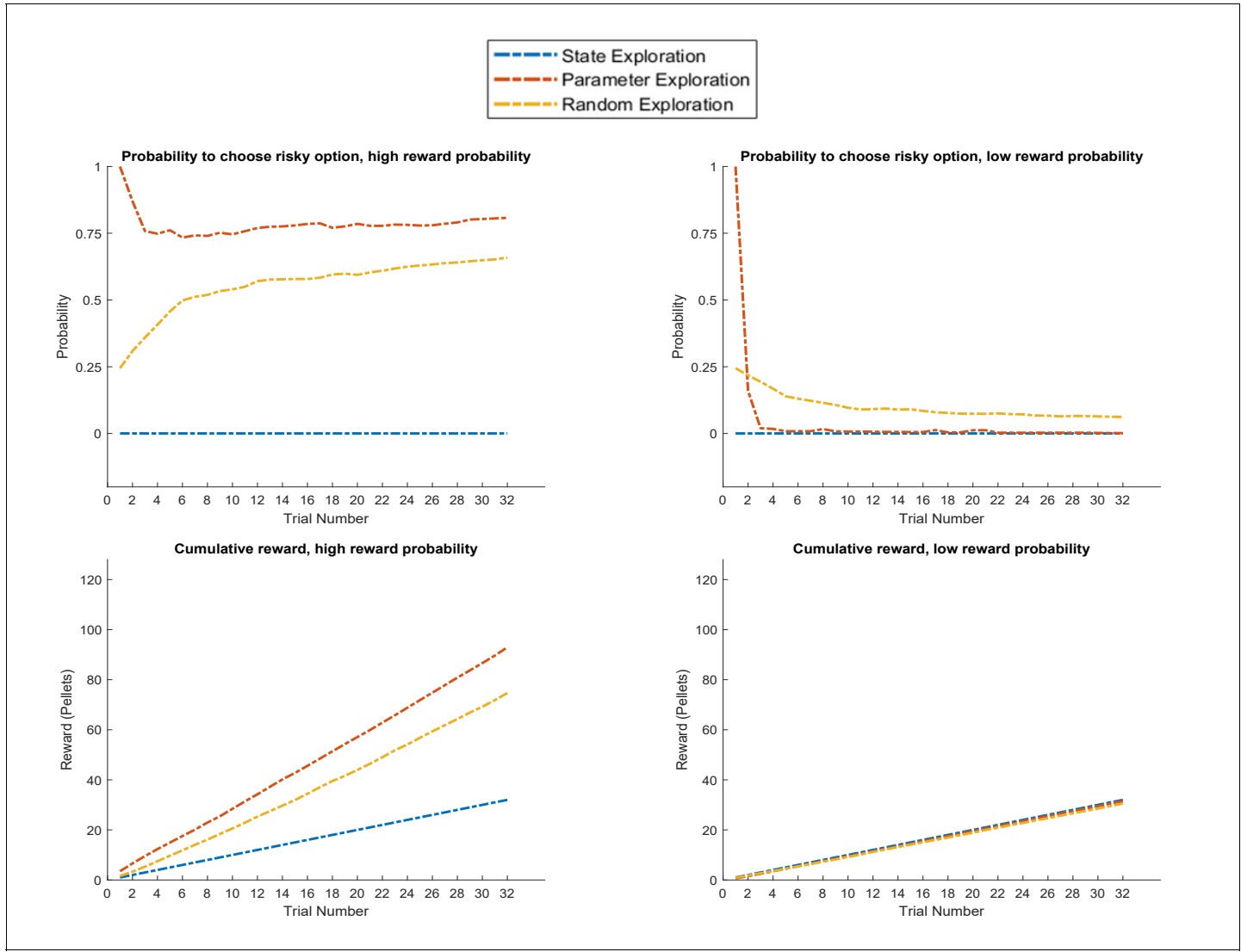

**Figure 10.** Response profiles of a 'state exploration', 'parameter exploration' and 'random exploration' agent in a task that requires learning. In the task described at the top of *Figure 2*, only the 'parameter exploration' agent (no state exploration) flexibly adapts to the current reward statistics, whilst the 'state exploration' (no parameter learning) agent fails to form a representation of the task statistics. Upper panel: probability for each of the three agents to choose the risky option if it is associated with a high (left, 85%) or low (right, 15%) reward probability. Lower panel: average cumulative reward (measured in pellets, where low reward = one pellet and high reward = four pellets) in 100 simulated experiments in a high (left) and low (right) reward probability setting, indicating an advantage for the 'parameter exploration' agent when the risky option is associated with a high reward probability.
DOI: https://doi.org/10.7554/eLife.41703.011

aversive for a pure *active inference* agent, because they are associated with a high entropy (ambiguity) in their mapping to underlying hidden states, which an *active inference* agent is compelled to *minimise*. Consequently, it will always sample the safe option in this task. This induces a performance pattern in which the 'parameter exploration' agent is superior to the other two agents if the reward probability in the risky option is high, but not if the reward probability is low and the best course of action is to sample the safe option (left and right lower panel of *Figure 10*).

In situations where state exploration is a necessary means for good performance, we should expect a state exploration agent to outperform the other two. *Figure 11* compares the three agents in the task introduced in *Figure 8*, where the current context (high or low reward probability in the risky option) changes unpredictably on a trial-by-trial basis but can be inferred from sampling a cue that signifies the current context. This illustrates the opposite situation to *Figure 10*: here, the 'state exploration' agent clearly outperforms the 'parameter exploration' and 'random exploration' agent. Importantly, this illustrates that when the context changes randomly, there is no knowledge that could be carried over from one trial to the next. Thus, *active learning*, which focuses on making observations that allow to transfer insights from one trial to the next, will be ineffective. In contrast, *active inference*, which focuses on making observations that allow for precise inference about the current hidden state (context) at a trial, provides an effective solution to this problem (cf. *Figure 7*), such that this agent always correctly infers the current context of a trial and, in consequence, whether to sample the safe or risky option.

*Figure 12* compares the three agents in the task introduced in *Figure 8*, which has the same design as the previous example but now with a stable (high or low) reward context across the entire experiment. This task can be solved with both active learning and active inference. The active learning agent has a high bias for sampling the risky option in the beginning of the experiment, and will

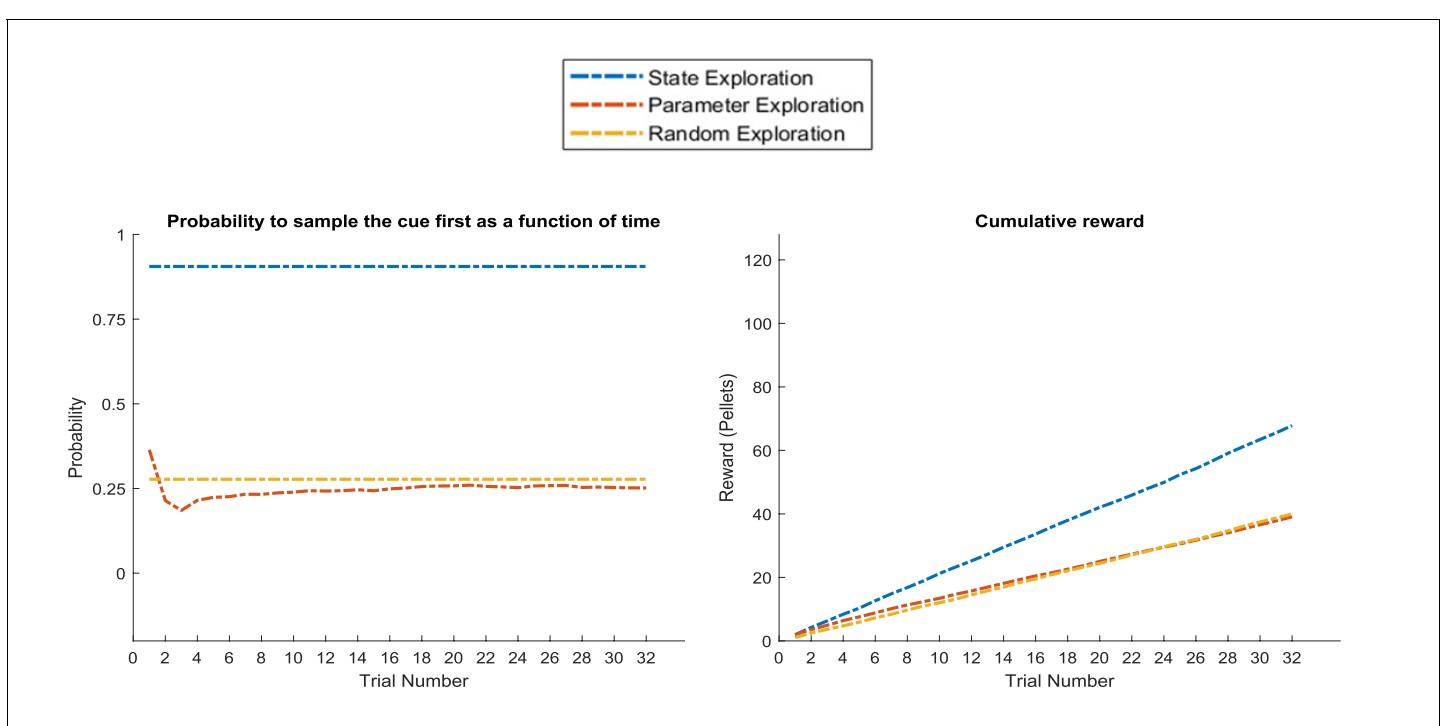

**Figure 11.** Response profiles of a 'state exploration', 'parameter exploration' and 'random exploration' agent in a task that requires inference. In the problem introduced in *Figure 7*, where an agent can infer the randomly changing context from a cue, 'parameter exploration' will be ineffective, because there is no insight that could be transferred from one trial to the next. 'State exploration', in contrast, provides an effective solution to this task, because it allows an agent to infer the current context on a trial-by-trial basis. Left panel: probability to choose the informative cue at the beginning of a trial. This shows that only the 'state exploration' agent correctly infers that it has to sample the cue at the beginning of every trial to adjust its behaviour to the current context (defined as a high or low reward probability in the risky option). Consequently, it outperforms the 'parameter exploration' and 'random exploration' agent in its cumulative earnings in this task (right panel).
DOI: https://doi.org/10.7554/eLife.41703.012

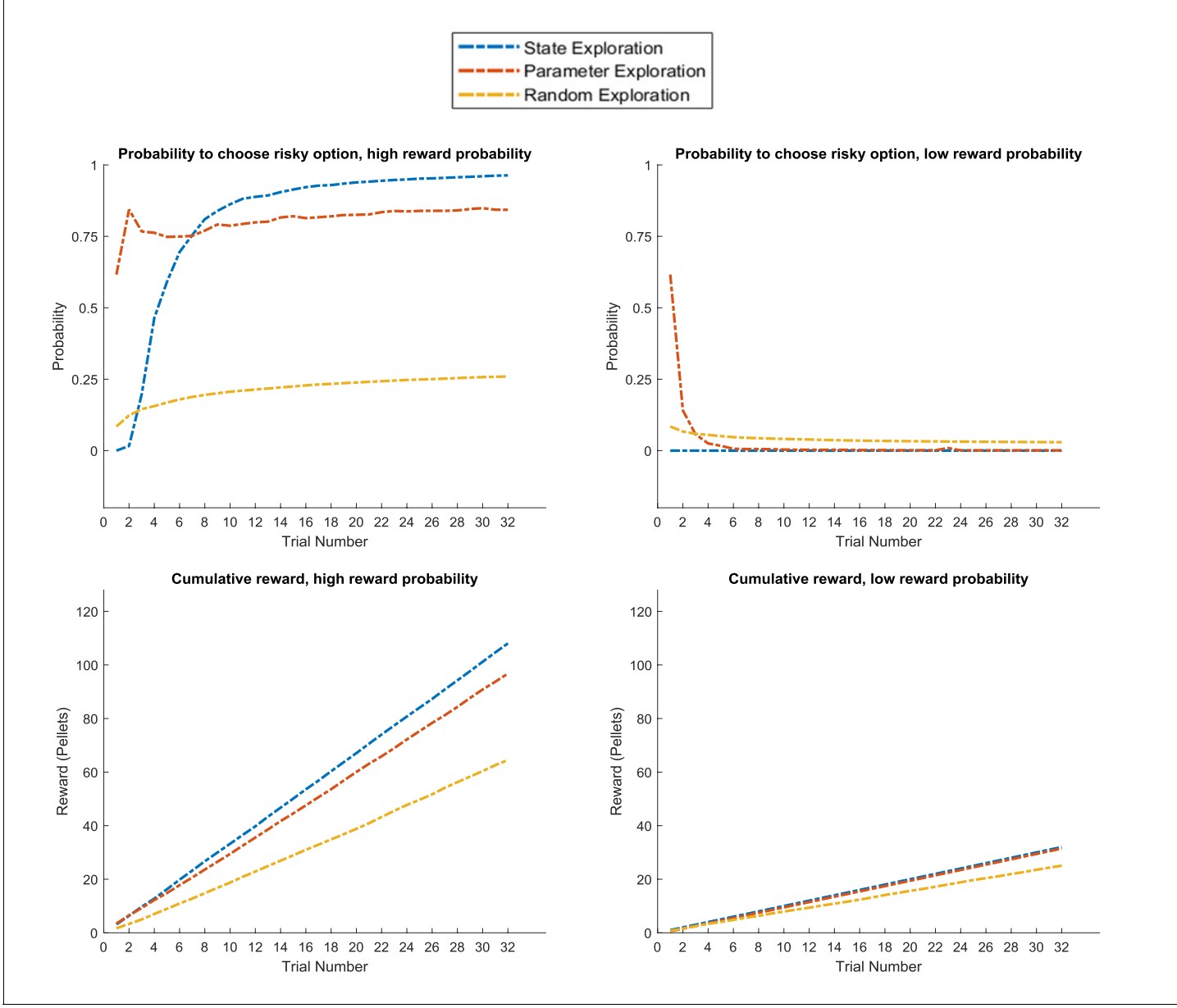

**Figure 12.** Response profiles of a 'state exploration', 'parameter exploration' and 'random exploration' agent in a task that requires learning or inference. Same problem as in *Figure 11*, but now with a stable high or low reward context (as in *Figure 8B*). This task can be solved by either sampling the risky option to learn about its reward statistics ('parameter exploration'), or sampling the cue to learn about the current context and adjusting the prior over contexts due to constant feedback from the cue ('state estimation'). This can be seen in the response profiles in the upper panel, such that the 'parameter exploration' agent has a strong preference for sampling the uncertain risky option in the beginning of the trial (left and right), while the 'state exploration' agent only starts sampling the risky option at the beginning of the trial if it has sampled the cue several times before, which always indicates a high reward context (left, cf. *Figure 8* ). This leads to a similar performance level of these two agents as measured by the cumulative reward, which exceeds the performance of the 'random exploration' agent (lower panel).
DOI: https://doi.org/10.7554/eLife.41703.013

thus learn whether it is associated with a high or low reward probability. The active inference agent has a strong preference for sampling the cue in the beginning of the experiment, but can adjust its prior over the current context due to stable (high or low reward) feedback from the cue (as illustrated in *Figure 8*). Thus, both the 'state exploration' and 'parameter exploration' agent will clearly outperform the 'random exploration' agent.

In sum, we have outlined different types of goal-directed exploratory behaviour that emerge under a probabilistic account of behaviour. In tasks where there is no hidden state that can inform an agent about current reward contingencies, an active learning agent performing parameter exploration will outperform an active inference agent performing hidden state exploration. In contrast, if there is an informative hidden state that changes unpredictably, active inference outperforms active learning. Only if there is a stable hidden state for a longer period in a task, both active learning (by learning from observations) and active inference (by gathering information about the hidden state) will lead to adaptive behaviour. This illustrates an important difference between active learning and active inference. Active learning is most efficient over longer timescales if information remains relevant over trials, whereas active inference is most efficient over shorter timescales, when contingencies can change on a trial-by-trial basis.

This concludes our investigation of the different response profiles of parameter exploration and hidden state estimation in different tasks (but see appendix for further simulations on the effect of other parameters on these behaviours). Next, we explore how these types of goal-directed exploration relate to empirical results on information gain in animals. We refer to Appendix 2 for a discussion of other computational frameworks of curiosity and exploration and their relation to the computational architecture we have presented here.

## Behavioural and neuronal predictions

In this section, we will discuss key behavioural and neuronal predictions of active inference and active learning, serving two purposes. First, we present testable predictions for behaviour and the neuronal mechanisms of active learning and active inference. Second, we discuss empirical evidence in relation to these predictions. Despite using a 'rat' as an exemplar agent above, the model-based predictions reported here are not restricted to rodents. Consequently, we will discuss various predictions by drawing from the entire animal literature.

### Active inference and active learning in behaviour

While active inference and active learning provide a general and flexible architecture for inferring individual differences in behaviour (see appendix), it nevertheless makes specific predictions about the interplay of exploitative and exploratory behaviour. A key prediction is that information should have an *additive* effect in relation to reward (cf., *Figure 1* and *Equation 7*, Materials and methods section). This means that an agent's reward- and information-sensitivity can be manipulated separately. Importantly, there is an *implicit* weighting for the tendency towards exploitation and (goal-directed) exploration. This weighing is determined by two factors: the precision of prior preferences and the degree of uncertainty about the world. If there is a high degree of uncertainty in an agent's observation model or beliefs about the current state, there will be a strong motivation for (intrinsic) active learning or active inference, respectively. If an agent's preferences over outcomes are very precise, on the other hand, then the (extrinsic) 'realising preferences' component will have a stronger impact on policy selection. These precision and uncertainty effects are distinct from the precision of policy selection that determine an agent's randomness – akin to an inverse temperature parameter in softmax response rules.

The implicit weighting between (intrinsic) information and (extrinsic) reward predicts that an agent's information-seeking behaviour will not be directly informed by the agent's utilities (cf., *Yang et al., 2016*, Box 2); for example, by being more sensitive to information about highly rewarding options. This implies that states can become 'interesting' that are entirely 'uninteresting' from an extrinsic reward perspective. However, empirical evidence shows that highly rewarding options may be more *salient* than options that are associated with a low reward, which is an important interaction that we will discuss in more detail below (*Figure 13*).

Another prediction for behaviour concerns the interplay of exploration and exploitation over time. In the simulations above, an agent's preference distribution is assumed to be stable, whereas her uncertainty about the world changes. Consequently, goal-directed information-gain will be prevalent at the beginning of a trial, whereas exploitative behaviour and stochastic action selection remain constant throughout a task. Note that we have focused on a simple one-shot active learning or two-shot active inference task; however, the present framework also accommodates exploration-exploitation trade-offs for larger policy depths, based on the sum of the expected free energies

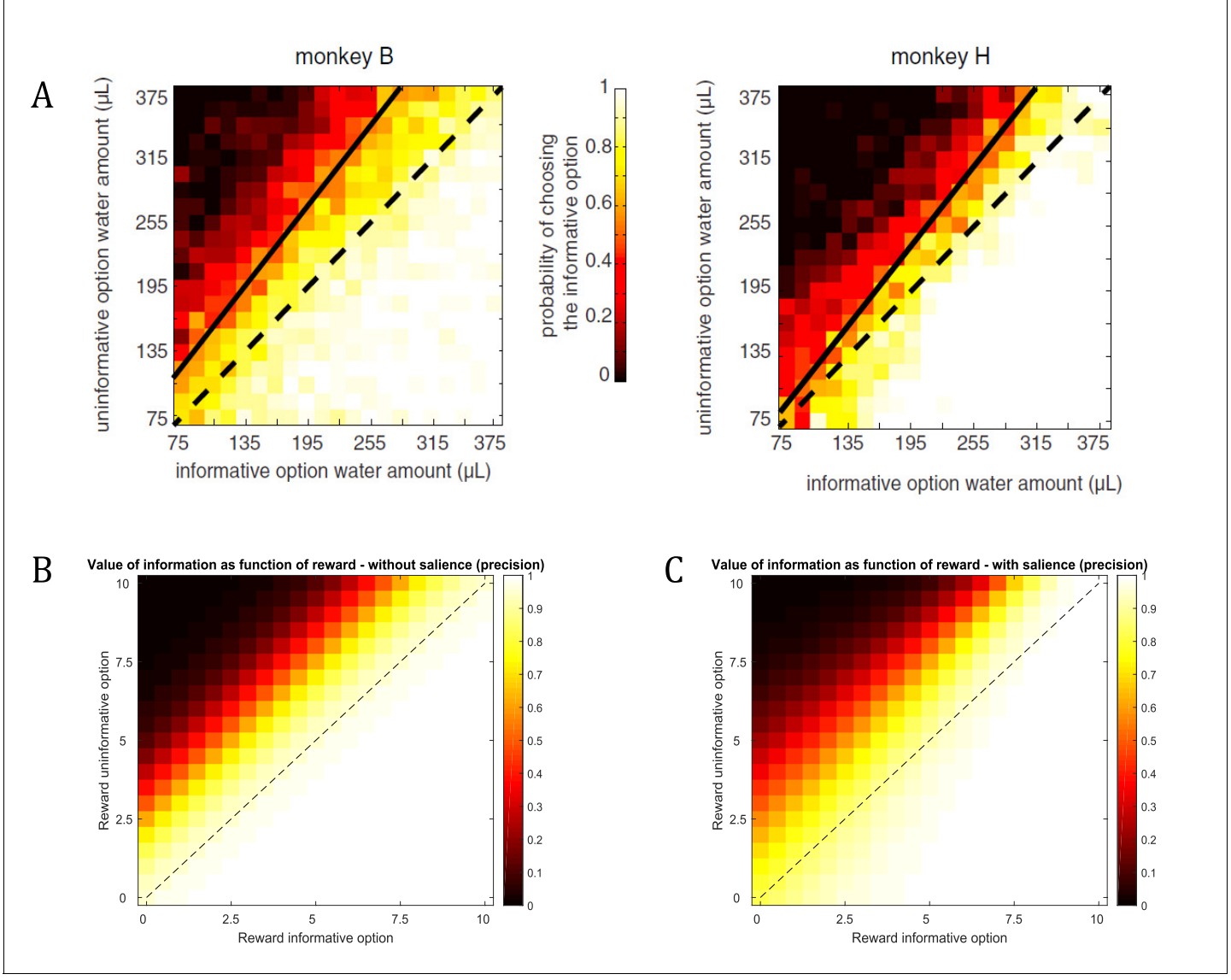

**Figure 13.** Dynamic relationship between reward and information. (A) Empirical findings from *Blanchard et al., 2015* suggest a modulatory effect of reward on the value of information. The higher the expected reward of the options, the more do monkeys prefer the option with the additional information about the reward identity during the delay period. For example, the preference for the informative option will be stronger if both options offer 315µL of water compared to both options offering 75µL of water. (B) Assuming a constant salience of different offer amounts (i.e. a constant precision of policy selection), active learning (and inference) predicts a preference for the informative option that is constant across different reward amounts (simulated from 0 to 10 pellets). That means that the preference for the informative option is the same when both options offer 1 or 10 pellets, for instance. (C) When taking a dynamic change of the precision of policy selection for different offer amounts into account (ranging parametrically between $\beta = 2$ for zero pellets in both offers and $\beta = 2^{-1}$ for 10 pellets in both offers), the simulated preferences match the empirical results from *Blanchard et al., 2015*. This highlights the importance of the interplay between the (extrinsic and intrinsic) values of options in active learning and active inference.

DOI: https://doi.org/10.7554/eLife.41703.014

over time (*Equation 4*, Materials and methods section). This will make an agent sensitive to large information (or reward) gains at later time-steps and not just the subsequent step (i.e., the agent will not be myopic).

Several lines of experimental work suggest that animals are sensitive to information gain, and assign a value to information that competes with a reward-based (extrinsic) value of an option.

Often, these behavioural tests are based on 'cue signalling tasks'. In these tasks, an animal chooses between two options followed by a reward after a delay. Crucially, in one of these options the outcome is signalled to the animal during the delay; such that the animal can resolve its uncertainty about the upcoming reward. This additional signal – during the delay – has no instrumental value and does not shorten the delay period. It is debatable to what extend this task truly reflects an explicit value of information in exploratory (curious) behaviour as simulated above (*Wang et al., 2018*); rather than just a change in the anticipation of rewards that may itself be attractive (*Iigaya et al., 2016*). Either way, however, this paradigm assesses an animal's preference for non-instrumental information, as opposed to pure exploitative behaviour. In the following, we use empirical paradigms inspired by the 'cue signalling task' to discuss behavioural and neuronal evidence for the encoding of 'information' implicit in active inference and learning.

Past work has shown that animals assign a value to gaining information about the outcome in such 'cue signalling tasks'. Pigeons appear to prefer (on average) a two pellet option over a 'safe' three pellet option, if the two pellet option includes an additional signal about the reward size during the delay period (*Zentall and Stagner, 2012*; *Zentall and Stagner, 2011*). Analogously, starlings show a preference for an option with a lower reward probability, if there is an informative cue in the delay period (*Vasconcelos et al., 2015*). This effect is stronger if the cue is shown shortly after the animal's choice, as opposed to close to the outcome delivery. The same – from an economic perspective 'suboptimal' or 'bounded rational' behaviour – has been found in rats (*Chow et al., 2017*), monkeys (*Blanchard et al., 2015*; *Bromberg-Martin and Hikosaka, 2009*; *Smith et al., 2017*), and humans (*Iigaya et al., 2016*).

These empirical results suggest that non-instrumental information provides an additional value to an option, in addition to its external reward. Importantly, this additional informative value can render an option more valuable even though its objective (economic or extrinsic) value is lower than alternative options. As previously noted, one central prediction from the computational framework presented here is that information provides an additive value to an option, which is evaluated alongside its extrinsic value. This resonates closely with the above results, where the signalling cue provides an additional value to an option and makes it more likely than alternative options that have higher extrinsic (economic) value.

However, the assumption that the value of an option reflects the linear sum of its extrinsic and intrinsic value may not always be true. Importantly, *Blanchard et al., 2015* have found that monkeys are *more* sensitive to information if the option has higher extrinsic value (see *Figure 13A*). Using water as reward, they report that "the value of information may have a multiplicative effect on the value of water amount, just as probability does in a conventional gambling task, time does in a discounting task, or effort does in an effort task''. That means that the more water was at stake for a given gamble, the more monkeys preferred to choose the option that included a signal during the delay period before receiving the outcome.

*Figure 13* reproduces the effects reported by Blanchard et al., based on an active learning agent (assuming that now there are two options with a 0.5 probability for obtaining a reward, but with an information bonus for one of the two options). We find a strong behavioural bias towards the informative option, confirming the experimentally observed information bias in monkeys. This is in line with the additive effect of information on the value of a policy as predicted by active learning. *Figure 13B* shows that this additive effect results in a constant preference for sampling the option that is associated with higher uncertainty, irrespective of the reward magnitude of the two options (simulated for 0 to 10 pellets). In other words, the agent will consistently prefer the uncertain option if the objective values are the same (diagonal of *Figure 13b*). This agent, however, is insensitive to the total amount of reward, such that it will exhibit the same preference for the informative option if both options offer 1 or 10 units of reward, for instance, which is in contradiction with Blanchard et al.'s empirical findings. Importantly, active learning and active inference can also account for an interaction between reward and information, as reported in the original results (shown in *Figure 13A*). A supra-additive effect (as seen by Blanchard et al.) is expressed when taking a dynamic nature of the precision of policy selection into account. Previous studies have shown that reward or information modulate attention (by acting as a salience signal), which in active learning and active inference is reflected by a change in the level of the precision in policy selection (*Friston et al., 2012*; *Feldman and Friston, 2010*; *Moran et al., 2013*; *Schwartenbeck et al., 2016*). This is in line with previous work on curiosity and exploration, where attention and salience

have been identified as mechanisms that modulate curiosity. For example, Kidd et al. have found that infants direct less attention towards information about overly simple or overly complex stimuli (*Kidd et al., 2012*; *Kidd et al., 2014*), suggesting that they 'implicitly decide to direct attention to maintain intermediate rates of information absorption' (*Kidd and Hayden, 2015*). Further, it has been shown that a neuronal effect for novelty critically depends on attention towards the reward-predicting feature of a stimulus, as opposed to when subjects had to make reward-unrelated judgements about stimuli (*Krebs et al., 2009*). Importantly, these results suggest that attention towards the rewarding properties of a stimulus modulate the effects of its informative value.

*Figure 13C* illustrates this point by assuming a change in the precision of policy selection as a function of the value of options (precision ranging parametrically from $\beta = 2^1$ for zero pellets in both options to $\beta = 2^{-1}$ for 10 pellets in both options). Assuming policy precision is itself optimised, our model expresses a remarkably similar pattern to that observed in Blanchard et al., confirming the supra-additive effect of information. Thus, from the perspective of active learning (and inference), the empirical observation of a modulatory effect of reward on information speaks to an interplay between the value of different options, which provide empirically testable behavioural and neuronal predictions (see open questions below). Note that in the simulations of *Figure 13C* we varied the (hyper-)prior ($\beta$ in *Figure 1*) on precision in analogy to our simulations above. An extensive body of work, however, investigates the time-sensitive updating of precision ($\gamma$ in *Figure 1*) itself by treating it as a hidden state that can be inferred (*FitzGerald et al., 2015*; *Friston et al., 2014*; *Friston et al., 2017a*), to which we will return below.

## Neuronal mechanisms of active inference and active learning

While the focus of this work is on the computational mechanisms of exploratory and curiosity-driven behaviour, the theoretical framework of active learning and inference also makes predictions about the neuronal encoding of information and ensuing curiosity. It is thereby crucial to understand how (i) (expected) intrinsic and extrinsic value are represented neuronally, and (ii) how their neuronal encoding allows the processing and *updating* of information during active sampling. In particular, we will focus on two key results about the neuronal basis of information-gain and curiosity that have been reported across different species; namely, the encoding of information in subcortical dopaminergic structures and the orbitofrontal cortex (OFC).

The OFC has been reported to encode relevant task variables (predictions) during reward-guided decision-making in an orthogonal manner, such as the expected reward and the reward probability of different options (*Padoa-Schioppa and Assad, 2006*; *Rudebeck et al., 2008*; *Rushworth et al., 2011*; *Stalnaker et al., 2018*; *Wilson et al., 2014b*). Most importantly, *Blanchard et al., 2015* have detected different populations of neurons in OFC that encode expected reward (water) and expected information. This implies that OFC neurons signal reward and information in an independent and not integrated way, such that OFC may serve as a kind of workshop that represents elements of reward that can guide choice but not a single domain general value signal" (*Kidd and Hayden, 2015*). This is an important observation, because exactly this form of neuronal representation is predicted by the construction of an additive value signal based on active inference and learning, and thus makes OFC a key candidate for the encoding of extrinsic and intrinsic value of different options as predicted under this framework (*Equation 7* in Materials and methods).

A second key candidate for the neuronal implementation of active inference and learning is the dopaminergic midbrain. Dopamine is known to play a key role in orchestrating the cost-benefit trade-off implicated in the active inference examples above (*Hauser et al., 2017*, see Appendix 1 for a more detailed discussion). In addition, dopamine neurons have been shown to encode 'information prediction errors' analogously to 'reward prediction errors' (*Montague et al., 1996*; *Schultz et al., 1997*). *Bromberg-Martin and Hikosaka (2009)* found that dopaminergic neurons signal the information content conveyed by an informative cue in a cue signalling task, just as they signal unexpected (omissions of) reward. Importantly, this suggests that these neurons did not differentiate between (extrinsic) reward and (intrinsic) information. Second, a more recent study has shown that dopamine neurons signal prediction errors in reward as well as sensory prediction errors about reward identity that are orthogonal to the reward magnitude (*Takahashi et al., 2017*). These results suggest that the sum firing of dopamine neurons may reflect a 'common currency' for prediction errors about task information and extrinsic reward. Similar signals in dopamine-rich midbrain

regions have been implicated in recent studies in humans (*Boorman et al., 2016*; *Iglesias et al., 2013*; *Nour et al., 2018*; *Schwartenbeck et al., 2016*).

These empirical observations above are closely aligned with the formalism of active inference and learning as illustrated here. In the setting of active inference and learning, the function of dopamine has been linked to the role of the precision of policy selection (*Friston et al., 2014*; *Friston et al., 2012*). The role of this precision is twofold: as illustrated above, the (hyper-)prior on precision ($\beta$ in *Figure 1*) reflects the overall level of randomness (goal directedness) in behaviour. Additionally, precision is updated on a trial-by-trial basis based on variational (approximate) inference (not discussed or simulated above). These variational precision updates have the form of a prediction error between the prior and posterior expected free energy of policies (as described in *Figure 1* or *Equation 7* in Materials and methods, see *Friston et al. (2017a)* for a detailed treatment and derivation of these variational updates). In other words, the predicted dopamine responses will reflect the difference between the expected (prior) and actual (posterior) reward, model update and knowledge about hidden states (i.e. the difference between prior and posterior beliefs about the value of a policy). These predictions for dopaminergic signals correspond closely with empirical results reported in (*Bromberg-Martin and Hikosaka, 2009*; *Takahashi et al., 2017*), since an 'information' or 'identity' prediction error results from the difference between prior and posterior beliefs about model parameters and hidden states.

Dopaminergic neurons are a key target area of OFC, and it has been hypothesised that the prediction error signal in dopamine neurons may critically rely on OFC input (*Kidd and Hayden, 2015*; *Takahashi et al., 2011*), and that error-driven learning critically depends on the interplay between those two regions (*Jones et al., 2012*; *Takahashi et al., 2009*). Intriguingly, these speculations resonate with the fact that it required both the (intrinsic and extrinsic) value of options as well as the precision of policy selection in *Figure 13C* to reproduce the behavioural effect reported by Blanchard et al. (*Figure 13A*). Taken together, these simulations and empirical results suggest that active inference and active learning may critically depend on a factorised (intrinsic and extrinsic) value representation in OFC and a unified update signal in dopaminergic nuclei.

It is important to note that, besides the encoding of information in the OFC and dopaminergic nuclei, the physiological basis of exploration and active sampling has also been associated with other neuronal mechanisms. There has been much recent interest in the neuronal basis of active sensing in animals, such as whisking (*Bush et al., 2016*; *Campagner et al., 2016*; *Grant et al., 2009*; *Ranade et al., 2013*; *Yang et al., 2016*). Further, in humans exploratory choices and information gain have been correlated with activity in the insula and dorsal anterior cingulate cortex (*Blanchard and Gershman, 2018*; *Kolling et al., 2012*; *Muller et al., 2019*; *van Lieshout et al., 2018*) as well as the rostrolateral prefrontal cortex (*Badre et al., 2012*; *Daw et al., 2006*; *Ligneul et al., 2018*).

## Outstanding questions

While some of the key predictions from active learning and inference are closely aligned with empirical results, open questions remain, which we briefly outline below.

- It remains to be established to what extent active learning and active inference are mechanisms dissociable in behaviour and brain function, and to what degree these mechanisms make use of different cognitive and physiological resources. For instance, it is unclear whether the preference for advance information in the 'cue signalling task' can be attributed to active learning or active inference. One possible way to dissociate these processes is by investigating the time-course of information-seeking behaviour. Active learning predicts a decrease of exploratory behaviour over time, because the agent's uncertainty over its observation model decreases with accumulated sensory experience. Active inference does not necessarily predict a decrease in information seeking in tasks where there is no enduring context that could be learned (*Figure 7*). Under such circumstances, active inference predicts a constant preference for sampling a cue (which appears more in line with empirical results from the 'cue signalling task').
- Timing effects have been shown to modulate information gain behaviour in the 'cue signalling task', such as a stronger effect of an informative cue right after a choice – as opposed to closer in time to the presentation of the outcome (*Vasconcelos et al., 2015*). Our framework

currently makes no predictions about such temporal effects but could well be extended to accommodate such timing effects.

- Similarly, in the framework presented here one would not expect to see horizon effects in behaviour, such that information about the task becomes more valuable with an extended temporal horizon. This contrasts with empirical results on exploration (*Wilson et al., 2014a*). It will be important for future work to accommodate such effects, for example, in terms of an (inverse) discounting parameter for information.
- Active inference and active learning predict a factorised value representation in its neuronal encoding, such as in OFC neurons. Importantly, in a task where animals can gain information about models and states, 'value' neurons should exhibit a factorised representation for the expected information about model parameters, states and reward. This is in line with but goes beyond a factorised representation of information and value as reported elsewhere (*Blanchard et al., 2015*).
- If the value of a policy is constituted by potential information about models, states and reward, then we would expect to see a prediction error for these three quantities. For example, dopamine neurons should signal the difference between expected and actual reward, but also between the expected and actual information about model parameters and hidden states (in line with *Takahashi et al., 2017*), for instance). Further, given that information about models and states as well as reward are measured in a common currency (i.e., expected free energy), one would expect to see trade-offs between those quantities. For example, one would expect a positive dopaminergic response for a mildly aversive but highly informative outcome.
- Our simulations predict that the supra-additive effect found by Blanchard et al., depends on the interplay between dopaminergic neurons and neurons in the OFC. This implies that if one disrupts dopaminergic input after extensively training an animal on the task, OFC neurons should reflect an additive (*Figure 13B*) but not supra-additive (*Figure 13C*) effect of reward and information.

Learning about the structure of a task, besides maximising extrinsic reward, likely depends on a yet unknown integrated circuit of neuronal systems. For example, it has been shown that representations of task structure in OFC critically depend on input from the ventral subiculum (*Wikenheiser et al., 2017*), but the neuronal encoding that underlies structure learning in the hippocampal formation, OFC and neuromodulatory systems (and their interaction) is largely unknown.

## Discussion

We have illustrated the emergence of active inference and active learning when casting choice behaviour as probabilistic inference. Under the assumption that behaviour maximises model evidence or (equivalently) minimises surprise over future outcomes, this implies that choice behaviour will reflect a tendency to fulfil preferences and maximise utility, but also to minimise uncertainty about the current state of the environment and the relevant task contingencies. Whilst the tendency to fulfil ones preferences reflects exploitative behaviour, uncertainty reduction induces exploratory behaviour. We have contrasted such 'goal-directed' exploratory behaviour with 'random' exploration caused by imprecise and stochastic behaviour that is unrelated to an agent's uncertainty about the world.

This perspective makes specific predictions for behaviour. In particular, it introduces a distinction between the uncertainty about current states, which can be resolved by active inference, and uncertainty about model parameters, which can be resolved via active learning. Both uncertainties motivate goal-directed exploration but make different predictions for actual decision-making. Minimising the uncertainty over hidden states predicts that agents will seek observations from which there is a clear and precise mapping to the underlying hidden state, such as moving to a vantage point to infer the location of prey or sampling a cue that allows to infer the current context. Importantly, this sort of uncertainty reduction depends on a particular representation of the structure of the task and a particular parameterisation of that representation, which allows an agent to assess the mapping from observations to hidden states. We argue that agents are also driven by minimising the uncertainty about this parameterisation itself, as illustrated in the first simulations on 'parameter exploration'. Minimising the uncertainty over model parameters can even result in behaviour that conflicts with minimising the uncertainty over hidden states – in situations where agents try to sample options that are associated with high ambiguity but also with high novelty or information gain.

Consequently, a key prediction for behaviour is that the uncertainty about contingencies will modulate the effect of uncertainty about hidden states on behaviour. An option will be very interesting (i.e. informative) if its outcomes are ambiguous due to high uncertainty about the mapping from this option to possible outcomes, but the same option will be highly aversive if the agent is very certain that it leads to ambiguous outcomes.

In other instances, 'model parameter exploration' and 'hidden state exploration' can motivate similar types of behaviour. Our simulations, however, highlight an important conceptual distinction between active learning and active inference in their respective time courses. As mentioned above, it is possible to cast our 'hidden state exploration' example as an active learning problem, if we assume that the current context is stable enough to be learned over time. A key requirement for *learning* the context, however, is that it is possible to carry information from one trial to the next. If this continuity is broken, for example by changing the context randomly on every trial, the agent has to rely on active inference in order to gain information about the task. Thus, our framework predicts that active learning will be particularly useful if there are stable regularities or rules in the world that can be learned. Active inference, on the other hand, will be useful if behaviour has to adapt to trial-by-trial changes. In other tasks, active inference and active learning may interact, such as by learning about specific contingencies within a particular context. For example, imagine your favourite craft beer brewery introduces a novel beer based on the flavour of coffee and oranges. This might present a suitable instance for actively learning about the parameterisation of your preferences for coffee and orange flavoured beer, resulting in a large curiosity-bonus for this choice. However, you might be aware that you have a strong preference for Lager over Stout. Consequently, it might be useful to actively infer the hidden state of the novel beverage by asking the bartender what sort of beer you will obtain before placing your curiosity-driven order.

These considerations highlight the distinction between 'goal-directed' exploratory behaviour in the form of minimising uncertainty about hidden states or model parameters, 'random' exploratory (i.e., imprecise) behaviour and exploitative decision-making. The trade-off between these behavioural tendencies is governed by their relative precision. For example, if an agent strongly prefers one particular outcome over all other outcomes, she will display predominantly exploitative behaviour with the aim of attaining this outcome. In contrast, if there is one option that is associated with very high uncertainty about its mappings to outcomes, behaviour will be dominated by sampling that option until its associated uncertainty is resolved. Our simulations also illustrate that random exploration becomes adaptive if active learning or active inference is broken (or impossible). If the uncertainty about model parameters and hidden states fails to inform behaviour, the only way to learn about the world is through a higher degree of random sampling of different options. Our simulations have shown that this is the only way to (slowly and inefficiently) learn about the advantage of novel options in the absence of goal-directed exploratory behaviour. Further, it is important to note that these types of exploration themselves depend on a model of the task, such as an observation model or a model of the transitions between states. It will be a key challenge for future work to understand how agents build and compare these models in the first place, which provide the basis for inference and learning.

The distinction between active learning and active inference resonates with previous accounts of minimising different types of uncertainty, for example, the difference between unexpected and expected uncertainty (*Yu and Dayan, 2005*). However, the distinction between active learning and active inference emphasises the difference between uncertainty about model parameters and uncertainty about (hidden) states. This distinction can also be thought of as different *modes* of addressing different types of uncertainties. For example, an agent could reduce her unexpected uncertainty ('is the reward probability 50% or 99%?') via actively learning the 'reward parameterisation' of an option or via actively inferring the 'reward state'. Whilst active inference can often be a faster and more efficient way of reducing uncertainty, it also requires additional (structural) knowledge about the task, for example, that there is either a 50% or 99% 'reward probability state' but nothing in-between. Likewise, an agent can arrive at an accurate estimate of the expected uncertainty of an option ('there is a 50% chance that there will be a reward') via active learning and accumulating evidence that the true 'reward parameter' is 0.5, or via active inference and forming a precise belief that the current 'reward state' is 0.5.

This framework also promises a refined understanding of goal-directed cognitive deficits within the spectrum of neuropsychiatric disorders and accompanying animal models. For example,

individuals diagnosed with Schizophrenia and schizoaffective disorders are reported to suffer from cognitive deficits associated with both active inference and active learning (*Koch et al., 2010*; *Morris et al., 2008*; *Waltz et al., 2007*; *Weickert et al., 2009*). However, a specific characterisation of how diagnosed individuals differ in their information gain during adaptive behaviour has remained elusive. In a scenario where an individual suffers predominantly from working memory deficits or contextual integration failures, active inference may become impaired, resulting in reduced adaption to task contingencies. On the other hand, reduced efficiency in trial and error learning and attentional outcome deficits may speak to impairments predominantly in active learning. Whilst active inference and active learning may not always be entirely independent, this framework proposes the translation of individual performance into measurable parameters that reflect these different strategies in behaviour. A successful empirical dissociation of these behavioural phenotypes promises a refined assessment of cognitive deficits due to a more accurate understanding of the underlying neuropathological mechanisms, resulting in improved differentiability and predictability of cognitive dysfunction in individual subjects.

In summary, we have highlighted the distinction between learning about the world as a consequence of random or imprecise behaviour ('random exploration') and goal-directed uncertainty reduction. Further, we have shown how these types of behaviour arise when casting behaviour as probabilistic inference. We have identified two types of goal-directed exploratory behaviour, namely *active learning* that reduces the uncertainty that relates to the parameterisation of an agent's generative model of the world, and *active inference* that reduces uncertainty about hidden states in the world given an agent's generative model. This former type of uncertainty-reduction will compel an agent to sample novel contingencies that enable learning about the true mappings and thus induce 'model parameter exploration'. The latter type of uncertainty-reduction about hidden states motivates agents to sample salient observations that allow for precise, unambiguous inference about the current state, thus performing 'hidden state exploration'. We have shown that this distinction makes relevant predictions for the predominance of different types of exploration in different tasks depending on whether active learning or active inference is more adaptive. This will be critical for understanding the different motives underlying curiosity and information-seeking in animals and artificial intelligence, and provides mechanistic insight into suboptimal choice behaviour arising from broken *active inference* or *active learning*.

## Materials and methods

The generative model illustrated in *Figure 1* implies that outcomes (observations) are generated in the following way: first, a policy is selected using a softmax function of expected free energy for each policy (see below), which also depends on the agent's degree of randomness (precision) in behaviour. Sequences of hidden states are then generated based on the probability transitions specified by the selected policy. These hidden states then generate outcomes. State inference corresponds to inverting the generative model given a sequence of outcomes, while (parameter) learning corresponds to updating the mapping between hidden states and outcomes. Consequently, 'perception' corresponds to inferring (optimising) expectations about hidden causes with respect to variational free energy, while learning corresponds to accumulating concentration parameters. These variables constitute the sufficient statistics of the approximate posterior beliefs, denoted by the probability distribution $Q(s, \pi, A)$, where $s, \pi, A$ are the hidden or unknown variables (see below).

### Variational free energy and inference

In variational Bayesian inference (*model inversion*), one has to specify the form of an approximate posterior distribution. This form uses a mean field approximation, in which posterior beliefs are approximated by the product of marginal distributions over hidden causes. Here, this approximate posterior takes the following form:

$$Q(\tilde{s}, \pi, A) = Q(s_1|\pi)...Q(s_t|\pi)Q(\pi)Q(A)$$
$$Q(s_t|\pi) = Cat(s_t)$$
$$Q(\pi) = Cat(\pi)$$
$$Q(A) = Dir(A)$$

where Cat refers to a categorical distribution and Dir to a Dirichlet distribution. Note that we do not address belief updating about precision ($Q(\gamma)$) in our simulations, which have been discussed in detail in previous work (*FitzGerald et al., 2015*; *Friston et al., 2017b*).

Having specified a Markovian generative model and the approximate posterior, one can define the variational free energy and resulting update equations that are used to infer hidden causes, as well as the expected free energy over future states under policies, which defines the value of a policy.

Variational Bayesian inference implies that by minimising variational free energy with respect to the specified posterior $Q(x)$ over hidden causes $x$ (where $x = \{s, \pi, A\}$ in our example) we approximate the true posterior $P(x|\tilde{o})$:

$$Q(x) = \underset{Q(x)}{argmin} F \approx P(x|\tilde{o}) \tag{1}$$

There are several equivalent expressions for variational free energy: one is in terms of the entropy minus energy:

$$\begin{aligned} F &= \mathbb{E}_{Q(x)}[\ln Q(x) - \ln P(x, \tilde{o})] \\ &= \mathbb{E}_{Q(x)}[\ln Q(x) - \ln P(x|\tilde{o}) - \ln P(\tilde{o})] \\ &= D_{KL}[Q(x)||P(x|\tilde{o})] - \ln P(\tilde{o}) \end{aligned} \tag{2}$$

where $\tilde{o} = (o_1, \ldots, o_t)$ denotes observations up until the current time $t$. Because the (KL) divergence cannot be less than zero, the last equality means that free energy is minimised when the approximate posterior $Q(x)$ becomes the true posterior $P(x|\tilde{o})$. In this case, the variational free energy becomes the negative log evidence for the generative model.

Rewriting *Equation (2)* shows that variational free energy can also be written as

$$F = D_{KL}[Q(x)||P(x)] - \mathbb{E}_{Q(x)}[\ln P(\tilde{o}|x)] \tag{3}$$

This implies that minimising variational free energy maximises the expected likelihood of observations under the approximate posterior ('accuracy') whilst minimising the divergence between the approximate and true distribution over hidden causes ('complexity'). Having defined the objective function, the sufficient statistics encoding posterior beliefs can be updated by minimising variational free energy, as discussed in detail in (*Friston et al., 2017a*; see also appendix of *Parr and Friston, 2018* for the derivation of these updates). Given the focus of this paper, we will discuss inference on valuable policies in detail below.

As we have shown above, minimising free energy ensures that expectations about hidden causes are close to the true posterior over hidden causes, given observed outcomes. However, if we want to apply this notion to define the value of actions and policies, we need to consider potential *future* outcomes and states under a given policy. This can be achieved by making the log prior probability of a policy the (negative) free energy expected under that policy (*Friston et al., 2017b*):

$$\begin{aligned} P(\pi) &= \sigma(-\gamma \cdot G(\pi))) \\ G(\pi) &= \sum_{\tau} G(\pi, \tau) \end{aligned} \tag{4}$$

where $\tau$ refers to a time-step in the future, $\tau \in \{t+1, \ldots, T\}$ with $t$ reflecting the current time step. Note that the expected free energy over future states that determines the value of a policy resembles the expected value of future reward in reinforcement learning (*Sutton and Barto, 1998a*), although there is no discount parameter over future states. $\gamma$ reflects a precision parameter that governs an agent's goal-directedness and randomness in behaviour, parameterised by a gamma function with rate parameter $\beta$ (see *Figure 1*). Based on these beliefs about policies, agents sample an action, where the randomness of this sampling is governed by a precision parameter $\alpha$ (see below). Here, we simulate 'one-shot' experiments, in which there is no time-sensitive updating of precision, but we illustrate the role of the hyper-prior on precision ($\beta$) to simulate stochasticity or 'random exploration' in behaviour.

Using the same definition of free energy as in *Equation 1*), we can now define the expected free energy that defines the value of a policy. To do so, we need to make two changes to the definition

of the free energy in *Equation 1*). First, we need to define an approximate posterior conditioned on a policy. Second, given that we evaluate policies with respect to future observations that have not yet occurred, the expectation in *Equation 1*) needs to incorporate those future states (*Parr and Friston, 2018*). Consequently, we obtain $\tilde{Q} = Q(o_\tau, s_\tau, A|\pi)$, and by defining $\tilde{Q} = Q(s_\tau, A|\pi)P(o_\tau|s_\tau, A)$, we can write the free energy as (see *Solopchuck, 2018* for a step-by-step tutorial of this derivation):

$$G(\pi, \tau) = \mathbb{E}_{\tilde{Q}}[\ln Q(s_\tau, A|\pi) - \ln P(o_\tau, s_\tau, A|\pi)] \tag{5}$$

$$\begin{aligned}G(\pi, \tau) &= \mathbb{E}_{\tilde{Q}}[\ln Q(s_\tau, A|\pi) - \ln P(o_\tau, s_\tau, A|\pi)] \\ &= \mathbb{E}_{\tilde{Q}}[\ln Q(A) + \ln Q(s_\tau|\pi) - \ln P(A|s_\tau, o_\tau, \pi) - \ln P(s_\tau|o_\tau, \pi) - \ln P(o_\tau)] \\ &\approx \mathbb{E}_{\tilde{Q}}[\ln Q(A) + \ln Q(s_\tau|\pi) - \ln Q(A|s_\tau, o_\tau, \pi) - \ln Q(s_\tau|o_\tau, \pi) - \ln P(o_\tau)]\end{aligned} \tag{6}$$

And eventually (cf. *Friston et al., 2017a*; Equation 2.2):

$$G(\pi, \tau) = \underbrace{\mathbb{E}_{\tilde{Q}}[\ln Q(A) - \ln Q(A|s_\tau, o_\tau, \pi)]}_{\text{Model parameter exploration}} + \underbrace{\mathbb{E}_{\tilde{Q}}[\ln Q(s_\tau|\pi) - \ln Q(s_\tau|o_\tau, \pi)]}_{\text{Hidden state exploration}} - \underbrace{\mathbb{E}_{\tilde{Q}}[\ln P(o_\tau)]}_{\text{Realising preferences}} \tag{7}$$

This formulation of behaviour predicts that choices will be governed by three principles; namely, minimising uncertainty about model parameters (*parameter exploration* or *active learning*), minimising uncertainty about hidden states (*hidden state exploration* or *active inference*) and obtaining preferred outcomes (*realising preferences* or *goals*).

Preferences over outcomes are defined as prior (log-) expectations over outcomes (**c** in *Figure 1* and *Figure 2C*). Thus, policies become valuable if they minimise the deviation between expected and actual outcomes, which introduces the concept of surprise minimisation to choice behaviour. The focus of the present paper, however, is on the first two terms of the value of a policy.

The first term in the equation above reflects the mutual information between beliefs about *model parameters* before and after making a new observation and reflects *active learning* (cf., *Yang et al., 2016*). The notion of finding policies that maximise mutual information is equivalent to maximising (expected) Bayesian surprise (*Itti and Baldi, 2009*), where Bayesian surprise is the divergence between posterior and prior beliefs about hidden causes. Because mutual information cannot be less than zero, it disappears when the (predictive) posterior ceases to be informed by new observations. This means that 'active learning' will search out observations that resolve uncertainty about the world (e.g. foraging to resolve uncertainty about the reward probability of a risky option). However, when there is no posterior uncertainty – and the agent is confident about the structure of the world – there can be no further information gain and preferences over outcomes (i.e. rewards or utility) will dominate policy selection. This resolution of uncertainty is closely related to satisfying artificial curiosity (*Schmidhuber, 1991*; *Still and Precup, 2012*) and the 'value of information' (*Howard, 1966*). The second term of the value of a policy, on the other hand, reflects the mutual information of believes about *states* before and after making an observation and reflects *active inference*. This term quantifies how well agents can infer the underlying cause of an observation, and motivates agents to seek observations that decrease uncertainty about the current hidden state with respect to this mapping. Taken together, these two terms predict that policies will be preferred if they allow agents to optimise the parameterisation of their observation model and at the same time make observations that enable precise inference about the state of the world, given their observation model.

The posterior mapping from hidden states to outcomes (**A**) is parameterised as Dirichlet distribution, whose sufficient statistics are concentration parameters (*Friston et al., 2016*). These concentration parameters effectively reflect the (normalised) number of times a particular combination of states and outcomes has been encountered.

Actual updates of an agent's observation model (A-matrix) at time-point $t$ take place via updating these concentration parameters with respect to current observations and an individual learning rate $\eta$ (*Friston et al., 2016*):

$$\ln A = \psi(a_t) - \psi(a_0) \tag{8}$$

where $a_t$ reflects the update of the concentration parameters depending on the observed state-

outcome mapping at trial $t$, $a_t = a_{t-1} + \eta \cdot \sum_t o_t \otimes s_t$ ($\otimes$ is the cross-product), and $a_0$ reflects the (prior) concentration parameters at the beginning of the experiment, with $\psi$ referring to a psi- or digamma-function (i.e. a column-wise normalisation of concentration parameters). Note that $a$ refer to the concentration parameters specifying an agent's observation model via $P(A) = Dir(a)$. Effectively, *Equation (8)* implies that learning of the observation model takes place by counting the number of transitions from one particular hidden state to a particular outcome, modulated by an individual learning rate.

From the perspective of this paper, the two key terms that define the value of a policy are the opportunities for information gain pertaining to the mapping between hidden states and outcomes (first term), and about hidden states (second term). The former reflects an agent's reduction in uncertainty about model parameters, whilst the latter reflects an agent's reduction in uncertainty about hidden states. These two terms imply that policies will be preferred if they resolve uncertainty about the way in which hidden states generate outcomes ('model parameter exploration') and about the hidden states underlying observations ('hidden state exploration').

In addition to the precision (stochasticity) of policy selection, one can define a precision of action selection under inferred policies, $\alpha$: $P(a_t|\alpha) = \sigma(\alpha \cdot \ln P(a_t|Q(\pi)))$, where $Q(\pi)$ reflects the approximate posterior beliefs about policies $\pi$. In the above simulations, we have set $\alpha$ to 4 in the active learning simulations and to 16 in the active inference examples. Note that we have defined different values for $\alpha$ based on the difference in the number of available policies in these two examples, but the results of our simulations do not depend on particular values of $\alpha$. In the section on 'Comparing model parameter and hidden state exploration', we have set $\alpha$ to 8 for both the active learning and active inference agent to ensure comparability of the two agents, and we have set $\alpha$ to 1 to create a random exploration agent (to ensure consistency across the different simulated tasks).

## Code availability

The above simulations are based on routines available that are available as Matlab code in the SPM academic software: http://www.fil.ion.ucl.ac.uk/spm/ based on the epistemic learning demo in the DEM toolbox of SPM.

The exact simulations and figures of this paper can be reproduced based on code at the following github repository: https://github.com/schwartenbeckph/Mechanisms_Exploration_Paper using the Curiosity_Paper_Figures.m File (*Schwartenbeck, 2019a*; copy archived at https://github.com/elifesciences-publications/Mechanisms_Exploration_Paper).

These simulations are based on a broader tutorial on active inference and active learning that can be found at https://github.com/schwartenbeckph/CPC_ActiveInference2018 (*Schwartenbeck, 2019b*; copy archived at https://github.com/elifesciences-publications/CPC_ActiveInference2018). This tutorial includes important procedures for fitting such models to real behaviour – and evaluating parameter recovery (i.e. identifiability). This application goes beyond the scope of this paper but represents a crucial step in applying active inference and learning models to empirical data.

## Acknowledgements

We thank Rani Moran and Timothy Müller for very helpful comments on an earlier version of this manuscript. KJF is funded by the Wellcome Trust (Ref: 088130/Z/09/Z). TUH is supported by a Wellcome Sir Henry Dale Fellowship (211155/Z/18/Z), a grant from the Jacobs Foundation (2017-1261-04), the Medical Research Foundation, and a 2018 NARSAD Young Investigator grant (27023) from the Brain and Behavior Research Foundation. THBF is supported by a European Research Council (ERC) Starting Grant under the Horizon 2020 program (Grant Agreement 804701). The Max Planck UCL Centre is a joint initiative supported by UCL and the Max Planck Society. The Wellcome Centre for Human Neuroimaging is supported by core funding from the Wellcome Trust (203147/Z/16/Z).

## Additional information

### Funding

| Funder | Grant reference number | Author |
|---|---|---|
| Wellcome | 088130/Z/09/Z | Karl J Friston |
| Jacobs Foundation | | Tobias U Hauser |
| Wellcome | 211155/Z/18/Z | Tobias U Hauser |
| European Research Council | Starting Grant 804701 | Thomas HB FitzGerald |

The funders had no role in study design, data collection and interpretation, or the decision to submit the work for publication.

### Author contributions

Philipp Schwartenbeck, Conceptualization, Formal analysis, Investigation, Writing—original draft; Johannes Passecker, Conceptualization, Writing—original draft; Tobias U Hauser, Formal analysis, Investigation, Writing—original draft; Thomas HB FitzGerald, Conceptualization, Investigation, Writing—original draft; Martin Kronbichler, Conceptualization, Supervision, Investigation, Writing—original draft; Karl J Friston, Conceptualization, Resources, Formal analysis, Supervision, Investigation, Methodology, Writing—original draft

### Author ORCIDs

Philipp Schwartenbeck http://orcid.org/0000-0001-8943-9965
Johannes Passecker http://orcid.org/0000-0002-4366-2691
Tobias U Hauser http://orcid.org/0000-0002-7997-8137
Karl J Friston http://orcid.org/0000-0001-7984-8909

### Decision letter and Author response

Decision letter https://doi.org/10.7554/eLife.41703.026
Author response https://doi.org/10.7554/eLife.41703.027

## Additional files

### Supplementary files

• Transparent reporting form
DOI: https://doi.org/10.7554/eLife.41703.015

### Data availability

Our submission only contains theoretical and simulation work, there are no datasets that could be made available. All the routines that are used for these simulations are available in the academic software SPM (https://www.fil.ion.ucl.ac.uk/spm/, DEM toolbox, 'epistemic value' routine), as indicated and described in the main text of our manuscript. Further, all code and data to reproduce the specific simulations of this manuscript are publicly available on github (https://github.com/schwartenbeckph/Mechanisms_Exploration_Paper; copy archived at https://github.com/elifesciences-publications/Mechanisms_Exploration_Paper).

The following datasets were generated:

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

## Appendix 1

DOI: https://doi.org/10.7554/eLife.41703.016

### Effects of other model parameters

While our focus is on comparing 'state exploration' and 'model parameter exploration', that is *active inference* and *active learning*, respectively, it is important to note that the behaviour illustrated in our simulations also depends on other key parameters, which we consider in the following. In particular, we will address the role of an agent's (prior) parameterisation of its observation model, an agent's preferences over outcomes and the parameterisation of beliefs about hidden states. We will refer to the task with no cue and an uncertain reward probability in the risky option as the *learning task*, and to the task with a cue and an unknown hidden state that determines the reward statistics as the *inference task*.

Active inference and active learning provide a general and flexible framework for understanding individual variation in exploitative and exploratory behaviour. Animals can differ markedly in their risk preferences, learning speed or sensitivity to uncertainty. As we will show in this appendix, different risk preferences result from an agents' preference distribution over outcomes. A predominance of risk-averse or risk-seeking behaviour arises from highly precise prior preferences that either emphasize not forgoing a small reward or obtaining a large reward, respectively. Further, a flat preference distribution induces a diminished prevalence of reward-based, extrinsically motivated behaviour and renders the animal more sensitive to information gain. The tendency towards epistemic foraging is also determined by an agent's learning rate, which affects the time-course of intrinsically motivated behaviour, and prior confidence, which (in analogy with prior preferences), determines overall sensitivity to information. Inferring such individual differences in an animal's 'world model' requires careful fitting of choice behaviour and model comparisons given a particular task of interest (see 'code availability' section).

### Prior uncertainty over the observation model determines the value of information

In active learning, the key prior that determines an agent's sensitivity to information is specified by the concentration parameters of the agent's observation model (in *Figure 2*). The effects of these concentration parameters are illustrated in *Figure 6A*, which affords two key insights. First, an agent prefers the risky option if it beliefs that receiving a reward is likely (lower right corner of *Figure 6A*) but prefers the safe option if it beliefs that receiving a reward is unlikely (upper left corner of *Figure 6A*). Second, and more importantly from the perspective of active learning, the agent will prefer to sample the risky option if it has an *uncertain* uniform prior about the reward statistics (lower left corner of *Figure 6A*) but prefer the safe option if it has a *certain* uniform prior about receiving a reward (upper right corner of *Figure 6A*).

The agent's prior over its observation model, however, incorporates additional knowledge about the task structure, such as the mapping from the safe location to a small reward or the observations in the starting position. The role of this parameterisation is illustrated in *Appendix 1—figure 1*. *Appendix 1—figure 1A* illustrates an experiment where an agent has a very imprecise prior over the observation model, by specifying $a_0$ as

$$a_0 = \begin{bmatrix} 1/4 & 1/4 & 1/4 \\ 1/4 & 1/4 & 1/4 \\ 1/4 & 1/4 & 1/4 \\ 1/4 & 1/4 & 1/4 \end{bmatrix}$$

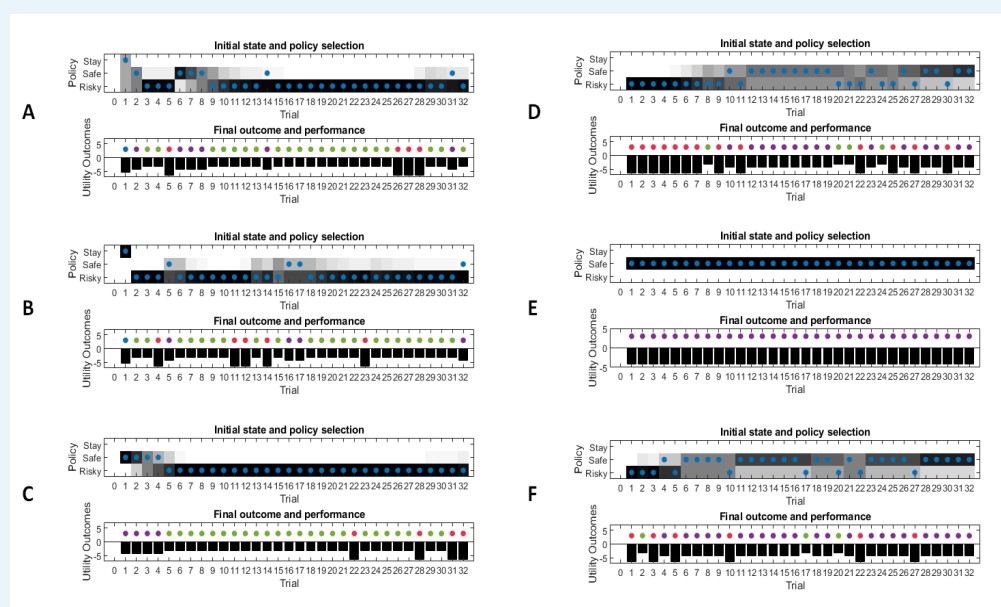

**Appendix 1—figure 1.** The agent's prior on the observation model determines the value of information of the risky option. Same setup as in *Figures 3–5*, but with different priors over the observation model. (**A**) Full uncertainty over an agent's observation model induces the same information value for all three options, whereas uncertainty over the mapping from the starting point or safe option induce a specific epistemic value for these options (**B and C**). These priors can also induce optimistic (**D**) or pessimistic (**E**) behaviour, based on a high (prior) reward expectation in a low reward probability (0.25) task or a low (prior) reward expectation in a high reward task (0.75), respectively. (**F**) A lower learning rate leads to slower learning about a low reward probability (0.25) in the risky option or, equivalently, to a longer dominance of information-seeking behaviour.

DOI: https://doi.org/10.7554/eLife.41703.017

This leads to a uniform probability for choosing to stay at the starting position, or choosing the safe or risky option at the first time-step. In this example, once the agent has sampled the starting position at the first time-step, it will then choose the safe or risky option at the second time-step, because now these two options have the highest information gain. Only after choosing the safe and risky option at time-step two and three, respectively, the agent starts to explore – and exploit in a more goal-directed manner. How long this pure information gain period lasts also depends on an agent's learning rate (see description of *Appendix 1—figure 1F* below). *Appendix 1—figure 1B and 1C* illustrate the same type of behaviour, but now with an imprecise mapping – from the starting position or the safe option – to different outcomes. Note that the information gain period lasts longer if there is uncertainty about the safe option compared to the starting position, because there is an additional exploitative motivation for choosing the safe option that delivers a small reward.

Figure *Appendix 1—figure 1D and 1E* illustrate the effects of non-uniform priors over reward statistics on active learning. *Appendix 1—figure 1D* corresponds to prior concentration parameters that reflect the upper right corner of *Figure 6* by specifying $a_0$ as

$$a_0 = \begin{bmatrix} 1 & 0 & 0 \\ 0 & 1 & 0 \\ 0 & 0 & 8 \\ 0 & 0 & 1/4 \end{bmatrix}$$

This induces a highly optimistic prior belief about obtaining a reward in the risky option. Simulating task performance under a low reward probability (p(reward)=0.25) shows that the agent samples the risky option several times before the observations overcome the (pathologic) optimism induced by the agent's prior. In contrast, *Appendix 1—figure 1E*

illustrates the opposite pattern (with the prior concentration parameter for receiving no reward in the risky option set to 2). This induces pessimistic behaviour – that results in a persistent preference for choosing the safe option. In this example, the agent never learns that it lives in an environment with a high reward probability (p(reward)=0.75), because it never samples the risky option, which would allow it to overcome its pessimistic prior beliefs.

Finally, *Appendix 1—figure 1F* illustrates a task where the agent learns only slowly from observations ($\eta = 0.05$ instead of 0.5 as in the simulations above, see *Equation 8*). Here, it takes the agent about the first third of the experiment to learn about the low reward probability (0.25) in the risky option. In other words, the value of information of the risky option dominates behaviour for a longer period than in the simulations illustrated in *Figures 3–5*.

## Prior preferences over outcomes determine risk preferences and the cost of information

Besides an agent's prior beliefs about the likelihood of outcomes, another key determinant of active learning and inference is an agent's prior preference over different outcomes. In the simulations above, we defined an agent's preferences as c = [0 2 4 -2], which reflect its preferences for the starting position, a small reward, a high reward, and no reward, respectively (see *Figure 2*). Note that in active learning and inference, these preferences are defined as log-probabilities over outcomes. For example, the above specification implies that the agent expects to end up in a high reward state $\exp(4) \approx 55$ times more than in the starting position ($\exp(0) = 1$). Consequently, the agent tries to infer behaviour that maximises these log-probabilities over observations, and thus minimises surprise.

Importantly, these preferences over outcomes determine an agent's risk preferences. *Appendix 1—figure 2* illustrates these effects in the active learning task, with a true underlying reward probability of 0.5 for the risky option. The ensuing behaviour shows that different risk preferences change the response profile for choosing the risky option over time.

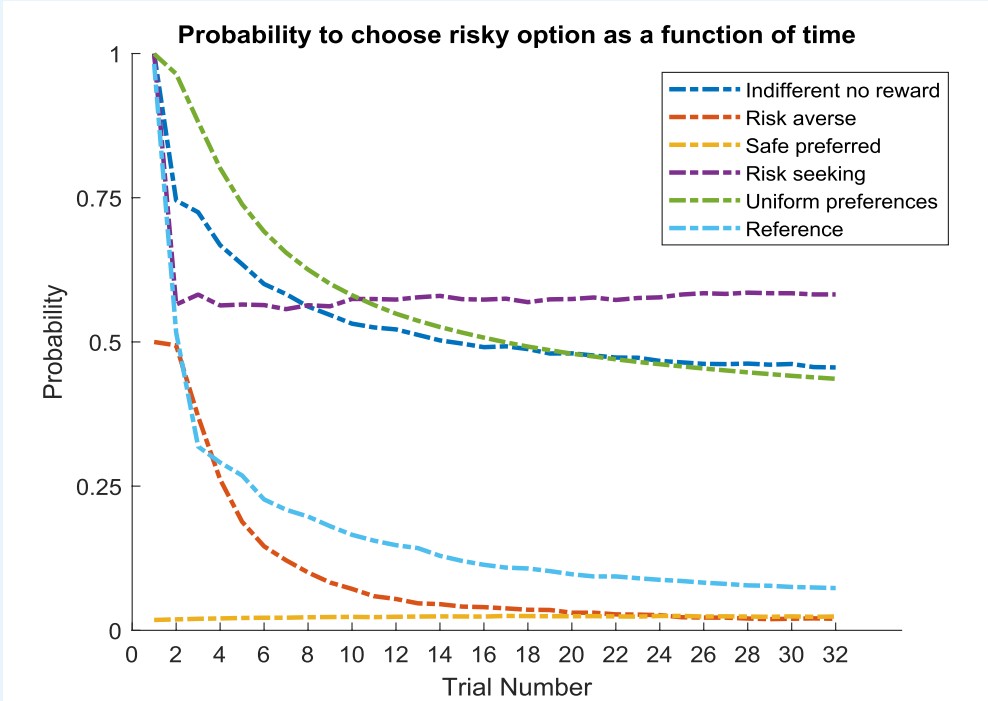

**Appendix 1—figure 2.** Prior preferences over outcomes determine an agent's risk preferences. Same setup as in *Figure 6A* but with varying preferences over outcomes, compared to the reference specification of c = [0 2 4 –2] in log-space used above (bright blue line), which

reflects an agent's preference for the starting position, the small safe reward, the high reward and no reward, respectively. Dark blue line reflects the same information seeking behaviour in the beginning of an experiment but less risk aversion in later trials by equating an agent's preference for obtaining no reward to staying in the safe position (c = [0 2 4 0]). Purple line reflects a risk seeking agent (c = [0 2 8 –2]) whereas red line reflects a risk averse agent (c = [0 2 4 –4]). Yellow line reflects an agent that has an equal preference for the high and low reward (c = [0 4 4 –2]), and consequently never chooses the risky option. Green line reflects an agent with flat preferences (c = [0 0 0 0]), which is purely driven by information gain until it converges on a stable probability for choosing the risky option. Time-course averaged over 1000 experiments.

DOI: https://doi.org/10.7554/eLife.41703.018

The above preference specification implies that – in log-space – obtaining no reward in the risky option is equivalent to losing the low reward of the safe option. *Appendix 1—figure 2* shows that a prior of c = [0 2 4 0] (dark blue line), which equates the preference for no reward with the starting position, makes the agent less risk averse compared to a prior of c = [0 2 4 –2] (bright blue line). In both parameterisations, behaviour is dominated initially by information gain, but converges to a higher probability for choosing the risky option over time, if the agent is less risk averse. Alternatively, a preference for no reward of –4 (red line) makes the agent much more risk (i.e., no reward) averse – with a zero probability of choosing the risky option, after the true reward probability is learned. This contrasts with a risk seeking agent that has a strong preference for obtaining a high reward (c = [0 2 8 –2], purple line). If the agent has the same preference for the small reward as for the high reward (c = [0 4 4 0], yellow line), the agent will never sample the risky option, because the extrinsic reward dominates policy selection. In contrast, if the agent has flat preferences over observations (c = [0 0 0 0], green line), behaviour will be dominated by information gain (in the risky option) in early trials and slowly converges towards a uniform preference for any of the three options (stay at starting point, safe or risky option).

Note that in the above parameterisation of an agent's preferences one would not necessarily distinguish between rewards (or their omission) and punishments, even though one could also introduce these as separate factors over outcomes (*Pezzulo et al., 2018*).

A central aspect of these simulations is that these preferences not only control an agent's risk preferences in active learning tasks, but also the cost of information as illustrated in the active inference task. This follows because log-preferences can be thought of in terms of cost (or, formally, counter-evidence to an agent's model). By navigating to preferred (expected) outcomes, the agent minimises cost (i.e., surprise).

This is illustrated in *Figure 8B*, where the agent infers a stable high reward context and shifts from sampling the cue at the beginning of a trial towards sampling the risky option immediately. There is no actual need to switch to sampling the risky option immediately, because there still are two time-steps available. However, given it has formed precise beliefs about the current task state, there is no additional value of information for sampling the cue. Sampling the cue is costly, because it is less preferred (expected) than obtaining a reward in the safe or risky option. This results from specifying an agent's preferences as c = [0 2 4 –2 0 0], reflecting its preferences for the starting location, the safe reward, the high reward, no reward and visiting the cue (signalling a high or low reward), respectively. This specification induces a cost for visiting the cue location as opposed to sampling the preferred option immediately. This cost for sampling the cue has to be matched by the information gain imparted by the cue, hence the shift from sampling the cue to sampling the risky option immediately at the beginning of a trial in *Figure 8B*.

*Appendix 1—figure 3* illustrates this point further. Compared to a neutral agent illustrated in *Appendix 1—figure 3B* (equivalent to *Figure 8B*), an agent that has a preference for sampling the cue (c = [0 2 4 –2 0.5 0.5], *Appendix 1—figure 3A*) will always go to the cue at the beginning of a trial, no matter how certain it is about the current context. In contrast, if the cue is even more costly (c = [0 2 4 –2 –0.5 –0.5], *Appendix 1—figure 3C*), the agent has a preference for leaving it as early as possible, but may have to return several times because it

has not fully resolved its uncertainty. Figure *Appendix 1—figure 4* illustrates these time-sensitive behaviours over multiple simulated experiments.

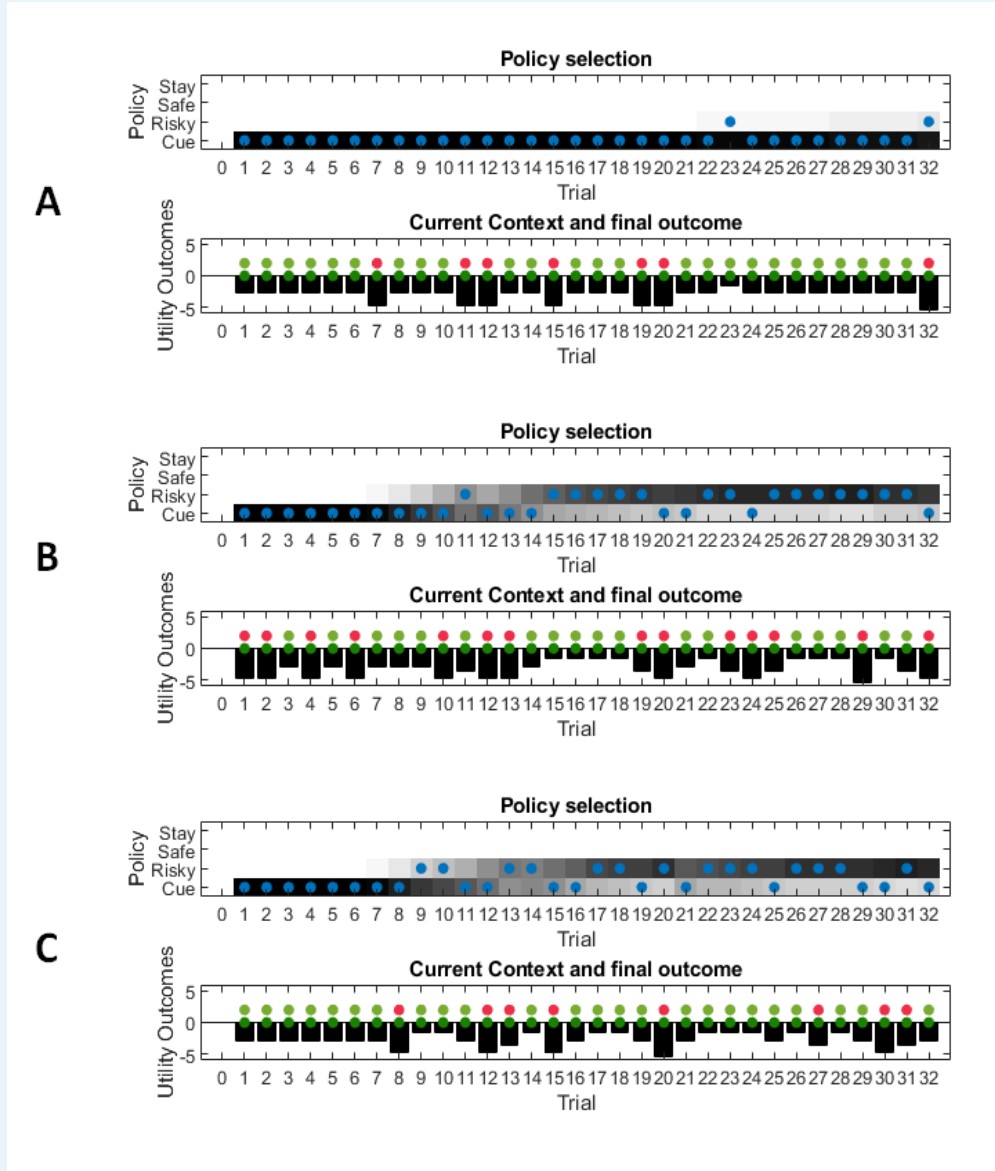

**Appendix 1—figure 3.** Prior preferences over outcomes determine the cost of sampling information in the active inference task – single experiment simulations. (**A**) An agent that prefers sampling the cue will continue to sample the cue at the beginning of a trial, even if its uncertainty about the hidden state has been resolved. (**B**) An agent with neutral preferences for the cue will switch to sampling the preferred (risky) option immediately once its uncertainty about the (high reward) hidden state is sufficiently resolved (equivalent to *Figure 8B*). (**C**) An agent with a negative preference for the cue will try to switch to the preferred option as quickly as possible, but may go back to sample the cue more often because its uncertainty has not been resolved sufficiently.

DOI: https://doi.org/10.7554/eLife.41703.019

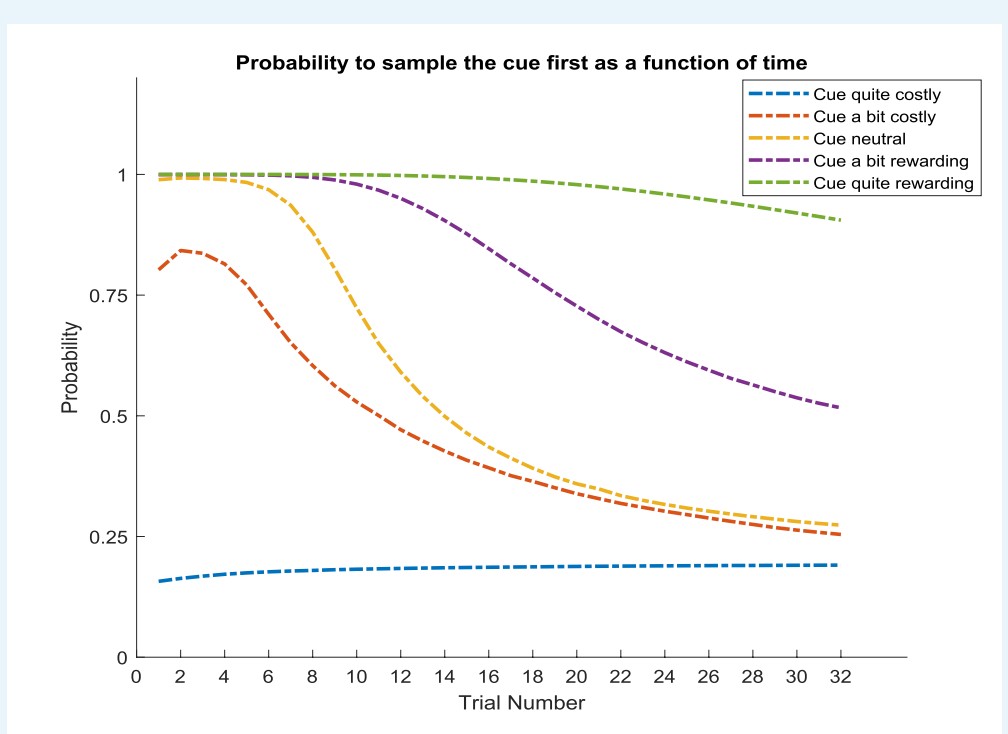

**Appendix 1—figure 4.** Prior preferences over outcomes determine the cost of information sampling in the active inference task – simulations over multiple experiments – simulations over 1000 experiments.. The yellow line reflects the agent introduced in *Figure 8B* with neutral preferences for the cue location, specified as c = [0 2 4 –2 0 0], which reflects its preferences for the starting location, safe reward, high reward, no reward and the cue location (signalling a high or low reward state), respectively. Agents with a slight preference for visiting the cue only slowly decrease their preference for sampling the cue at the beginning of a trial (purple: c = [0 2 4 –2 0.25 0.25]; green: c = [0 2 4 –2 0.5 0.5]). Agents with a negative preference for the cue location move away from sampling the cue quicker (red: c = [0 2 4 –2 −0.25 –0.25]; blue: c = [0 2 4 –2 −0.5 –0.5]). Time-course averaged over 1000 experiments.

DOI: https://doi.org/10.7554/eLife.41703.020

## Beliefs about hidden states in dynamic environments

Importantly, all simulations above are based on environments with a static structure, which allows agents to converge on one stable representation of the task over time. In reality, however, environments are volatile and change over time, such that new statistics have to be learned or novel hidden states have to be inferred. The simplest example of a dynamic environment is a reversal task, in which the true hidden state changes over time.

A detailed treatment of this issue goes beyond the scope of this paper. However, an important quality check for the present formulation is whether behaviour adapts to a changing environment. *Appendix 1—figure 5* and *6* illustrate adaptive behaviour in the active inference task. Here, we introduced a reversal of the hidden state after half of the experimental trials, such that the agent is in a high reward context in the first 16 trials followed by a low reward context in the next 16 trials (as indicated by a change from 'green' context to 'red' context in the second panel of *Appendix 1—figure 5A* and *Appendix 1—figure 5B*). *Appendix 1—figure 5A* shows that an active inference agent correctly infers the high reward context and moves slowly towards choosing the risky option immediately at the beginning of the trial (first time-step). After the switch to a low reward context in trial 16, however, the agent starts sampling the cue again. This behaviour is induced by a negative feedback in the risky option in trial 17 and is then reinforced by feedback from the cue, which now signals a low reward context. Thus, the active inference agent correctly infers whether the current context has

changed, and whether a recent increase in its uncertainty about the current context increases the value of information at the cue location again.

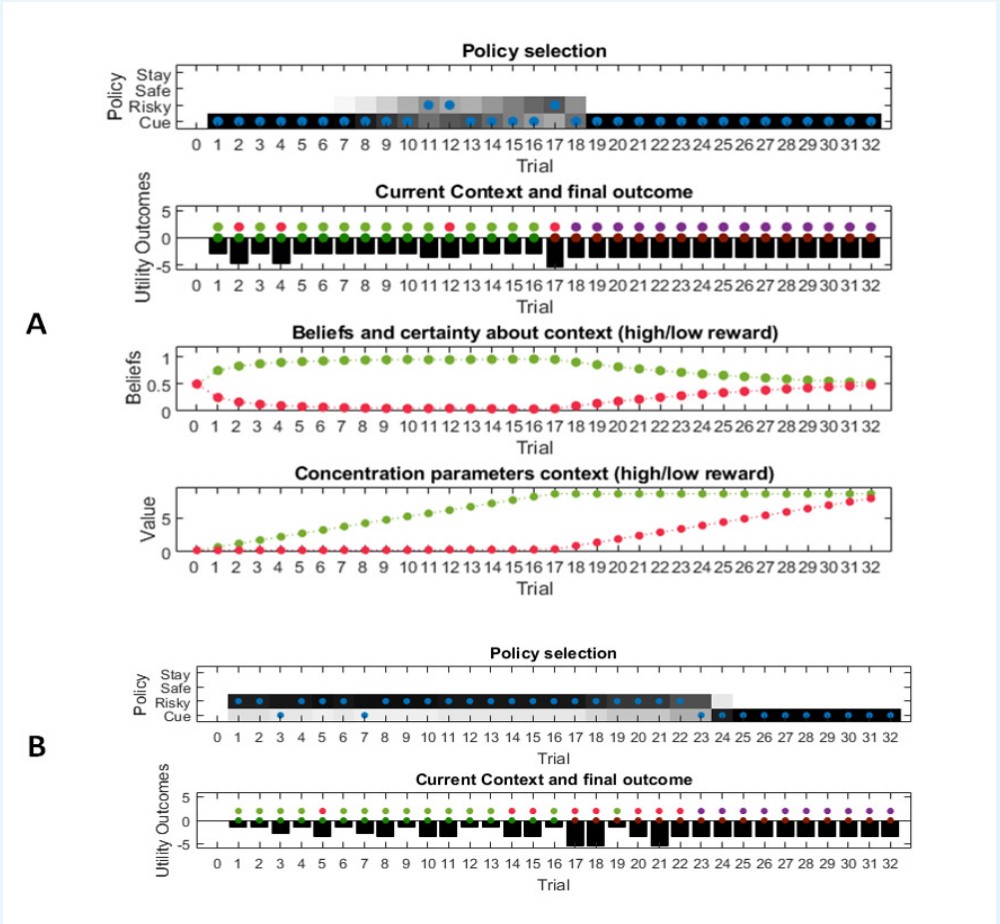

**Appendix 1—figure 5.** An active inference agent correctly infers a change in the environment – single experiment simulations. (**A**) An active inference agent correctly learns that it starts in a high reward environment, and slowly begins to sample the risky option at the beginning of a trial once it has inferred a high reward context. After a switch of context to a low reward environment in trial 16, however, the agent starts sampling the cue again. This is induced by a negative outcome at trial 17 and then further reinforced by a different context signalled by the cue. Thus, the active inference agent correctly infers when to start sampling information again as a function of its uncertainty about the world. (**B**) If an agent has optimistic prior expectations about being in a high reward context, it starts sampling the risky option immediately even at the beginning of the experiment, and it will take the agent longer to infer a switch of context in the second half of the experiment.

DOI: https://doi.org/10.7554/eLife.41703.021

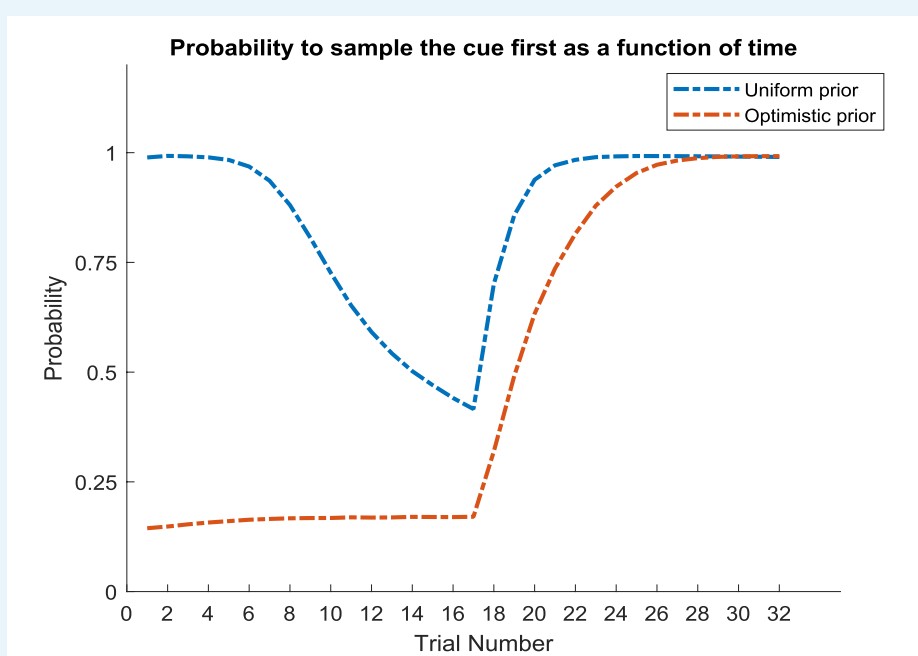

**Appendix 1—figure 6.** An active inference agent correctly infers a change in the environment – simulations over multiple experiments. Same agents as in *Appendix 1—figure 5* (**A** and **B**) simulated over 1000 experiments. The neutral active inference agent correctly infers a switch after the first half of the experiment and starts sampling the cue again. The optimistic active inference agent does not sample the cue in the first half of the experiment, but starts visiting the cue location once it has inferred that the context has changed.
DOI: https://doi.org/10.7554/eLife.41703.022

As introduced above, a central aspect of these simulations are an agent's beliefs about the current context, which can be parameterised with concentration parameters akin to an agent's observation model. These priors over beliefs about the context are specified in an agent's d-vector (see *Figure 7*). In *Appendix 1—figure 5A*, the d-vector reflects an uncertain uniform prior over being in one of the two contexts (d = [0.25 0.25 0 0 0 0 0 0]). *Appendix 1—figure 5B* illustrates an example where the agent has a strong prior expectation about starting in a high reward context (d = [8 0.25 0 0 0 0 0 0]). This motivates the agent to sample the risky option immediately without sampling the cue first, and only when it has sampled enough evidence in favour of a change of context (several trials after the actual change) it starts sampling the cue at the beginning of a trial again. *Appendix 1—figure 6* illustrates the time-course of behaviour for these two agents over multiple experiments.

## Appendix 2

DOI: https://doi.org/10.7554/eLife.41703.016

## Relationship to other computational approaches to exploration

There has been much recent interest in understanding the mechanisms of exploration and curiosity. As mentioned in the introduction, the simplest account of exploration has been cast in terms of $\epsilon$-greedy and softmax choice rules with an inverse temperature parameter (**Sutton and Barto, 1998b**), which governs the randomness in behaviour and thus the deviation from rational, exploitative choices. However, this account of exploration can be problematic because often it is unclear whether this truly captures (random) exploratory tendencies or just different forms of noise (e.g., Findling, Skvortsova, Dromnelle, Palminteri, & Wyart, 2018). In active inference and learning, the tendency towards randomness is captured by the (prior) precision of policy selection ($\beta$) as shown in the simulations in the main text. In the one-shot and two-shot tasks simulated here, this (prior) precision plays a very similar role to an inverse temperature parameter. In more sophisticated tasks with larger policy depths, precision itself will be updated (inferred) over time, implying that an agent's 'inverse temperature' has a time-sensitive Bayes optimal solution and the agent infers the 'best' level of randomness based on task events.

Two prominent examples of more sophisticated routines for goal-directed and random exploration are upper confidence bounds (UCB) and Thompson sampling, respectively. These computational frameworks have been discussed in much detail in previous work (**Gershman, 2018a**; **Gershman, 2018b**; **Schulz and Gershman, 2019**). The central idea of UCB is to add an additional value to an option that reflects its informative value. Similar to a novelty bonus (**Kakade and Dayan, 2002**), this additional value can reflect the number of times this option has been sampled previously (**Auer, 2002**; **Auer et al., 2002**) or the variance in an agent's beliefs about the value of this option (**Srinivas et al., 2010**). Thompson sampling, on the other hand, is a more sophisticated algorithm for random exploration. Here, the key idea is that agent's sample from beliefs about the reward statistics of different options, and then exploit (i.e., take the most valuable option) with respect to this sample. Thus, the uncertainty about the reward statistics guides the degree of randomness in behaviour, and consequently controls the degree of random exploration.

As shown in **Gershman (2018a)** and **Gershman (2018b)**, these two classes of algorithms have different effects on the exploitation-exploration trade-off. Uncertainty bonuses such as UCB add a value to sampling an option in addition to its expected value. This induces an intercept shift in the probability for sampling this option, as shown in **Appendix 2—figure 1** (left panel). The higher the uncertainty bonus, the higher is the agent's preference for sampling this option, even if its expected value is smaller than other options (in line with the additive aspects of Blanchard et al.'s findings). In contrast, random exploration, such as Thompson sampling, induces a change of the slope in the choice function, such that more randomness induces a flatter choice function (**Appendix 2—figure 1** right panel).

**Appendix 2—figure 1B** illustrates that these effects closely correspond to the effects of active learning or active inference and precision on behaviour. 'Model parameter exploration' or 'hidden state exploration' is defined as an additive term in the agent's value function (**Equation 7**, Materials and methods section) and can thus be seen as aspects of information-directed exploration. Thus, it induces an intercept shift analogously to UCB (displayed in **Appendix 2—figure 1B** left panel for 'model parameter exploration' in active learning, effects for active inference are analogous). Precision, or randomness, on the other hand, affects the slope of the agent's preference function (**Appendix 2—figure 1B**, right panel) and plays a very similar role to an inverse temperature parameter.

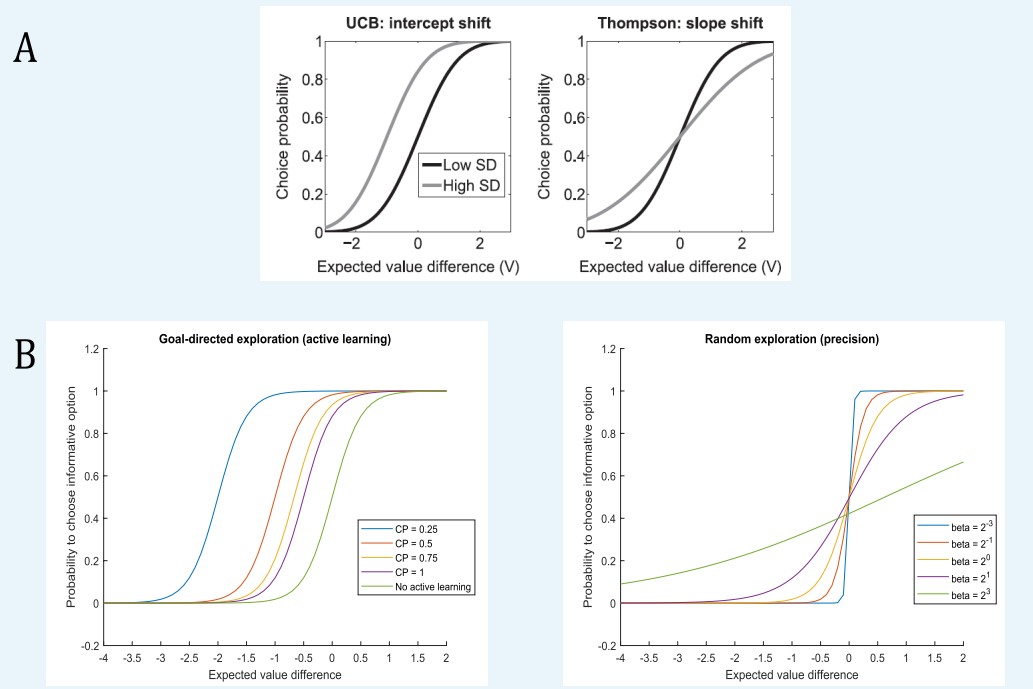

**Appendix 2—figure 1.** Effects of different algorithms for exploration on choice probabilities.
(**A**) *Left panel:* Algorithms based on an uncertainty bonus, such as UCB, change the intercept
in the probability for choosing the uncertain option, plotted as a function of the difference in
expected value between the uncertain option and an alternative option. *Right panel:*
Algorithms based on randomness, such as Thompson sampling, change the slope of the
choice probability, where an increase in randomness decreases the steepness of the choice
curve. Reproduced from (***Gershman, 2018a***) (**B**) *Left panel:* 'model parameter exploration' in
active inference acts as an uncertainty bonus and, analogously to UCB, changes the intercept
of the probability to sample an uncertain option as a function of the prior uncertainty over this
option. Different lines reflect different concentration parameters for the mapping to high or
no reward in the risky option (cf., ***Figure 2***, CP = concentration parameter). 'Hidden state
exploration' in active inference has analogous effects. *Right panel:* prior precision of policy
selection (*β*) affects the randomness of choice behaviour, and consequently the slope of the
choice function.Reprinted from ***Gershman (2018a)*** with permission from Elsevier. This panel is
not available under CC-BY and is exempt from the CC-BY 4.0 license.
DOI: https://doi.org/10.7554/eLife.41703.024

The recent interest in deep neural networks their application to more realistic tasks has led
to an increased focus on algorithms for exploration. This is motivated by a characteristic failure
of 'deep reinforcement learning' models in tasks that require goal-directed exploratory
behaviour, such as the Atari game Montezuma's revenge (***Burda et al., 2018b***; ***Mnih et al.,
2015***). These tasks are difficult because reward is sparse and many states are only visited
once, if at all, and thus motivate a more flexible definition of an intrinsic reward that guides
behaviour.

Much progress in this field is based on defining intrinsic motivation as the expected
learning progress in a given problem, such that agents 'plan to be surprised' (***Barto, 2013***;
***Burda et al., 2018a***; ***Luciw et al., 2013***; ***Oudeyer et al., 2007***; ***Oudeyer and Kaplan, 2007***;
***Schmidhuber, 1991***; ***Sun et al., 2011***; ***Itti and Baldi, 2009***). This is conceptually similar to
defining an uncertainty bonus for a given option. Given that these methods are often difficult
to apply to large-scale problems, other algorithms for novelty detection have been proposed.
Count-based methods use previous visits of a state as a measure for their novelty, with a
recent extension of pseudo-count methods to generalise these methods to more complicated
(non-tabular) problems (***Bellemare and Srinivasan, 2016***; ***Ostrovski and Bellemare, 2017***;

*Tang et al., 2016*). Other forms of novelty detection are based on (exemplar) classification methods (*Fu, 2017*) or 'network distillation', where the difference between a trained and a random network is used as a measure for novelty, based on how much an observation deviates from the training set (*Burda et al., 2018b*).

Perhaps closest to the framework presented here is the Variational Information Maximising Exploration algorithm (*Houthooft et al., 2016*), which is based on the idea of finding future trajectories (policies) that maximise the sum of entropy reduction in beliefs about environment dynamics. This means that agents should visit states that maximise the mutual information of prior and posterior beliefs about transition probabilities in the environment. Note that this follows a very similar logic to our treatment of active inference, except that it is applied to the agent's transition probabilities (B-matrix in *Figure 2*) rather than its observation model (A-matrix in *Figure 2*).

Note that, while many of the above algorithms have successfully introduced the notion of exploration and novelty to large-scale problems, a key motivation behind our proposed framework is understanding the generative mechanisms that underlie information gain and its trade-off with reward maximisation. How inference based on these generative mechanisms proceeds in more complicated tasks – where a representation of the feature or state space needs to be learned in the first place to build the A, B, c, and d-matrices of *Figure 2* and *7* – is an important question for future work.

