## [Decision Letter]

Thank you for sending your article entitled "Computational mechanisms of curiosity and goal-directed exploration" for peer review at *eLife*. Your article is being evaluated by Michael Frank as the Senior Editor and Reviewing Editor, and two reviewers.

Summary:

The authors explore the computational mechanisms of curiosity/information-seeking, focusing particularly on the distinction between two forms of goal-directed exploration. In particular they clarify through theoretical analysis and simulations the distinction between active inference (to reduce uncertainty about which world model is valid) and active learning (reducing uncertainty about the parameters of that model; e.g., outcome distributions or state transition probabilities), both within a framework of free energy reduction. The authors also outline circumstances in which active learning and/or active inference may be exposed as a function of the environment's structure and parameterization. Overall this is original work and the topics are timely.

Essential revisions:

Despite the strengths of the manuscript, the reviewers both had substantial reservations that would need to be addressed in a revision. The two reviewer points are consolidated below.

1) The reviewers were both concerned about the lack of connection of the analysis to empirical data that could be used to constrain or falsify it. They felt that the simulations are simple enough that there should be some data out there that could corroborate their model's predictions, and that for a life sciences (instead of purely computational) journal this is important.

- For example, the authors outline clear behavioral patterns that should be observed when an animal begins a simple t-maze task with risky options, or an experiment with a disambiguating cue. Are there existing data that could help support the model's claims here? Some evidence showing that rats preferentially sample from a risky option early, but switch to the low-risk option over time would certainly seem to be available in the corpus of animal behavior literature. In so doing, the reviewers also agreed that it would be important to compare your model to some alternatives, even if not doing formal model comparison with quantitative fitting, but at least qualitatively.

2) They also both agreed that there was very little connection to neural mechanisms despite various work that has been conducted in this domain. One reviewer noted "The decision to sample a cue to infer a hidden state involves cost-benefit tradeoffs relying on specific neural substrates but these are not mentioned at all. It also involves active sensing behaviors – such as whisking, about which a lot is known – but there is no real mention of any active sensing literature. The simulations in Figure 3 predict clear-cut risk aversion but there is no discussion of the fact that animals show mixtures of risk seeking and risk aversion when tested in these conditions. Active learning and active inference are associated with different neural substrates and behaviors but this is never mentioned when discussing the two mechanisms."

3) Technically, there were also a few important concerns that would have to be addressed:

- One concern was the flexibility of the free-energy framework, and how sensitive/robust it might be with respect to initial conditions and priors. For example, the agent in the first simulation takes an initialization as specified in Figure 2. Given that the agent is tasked with updating A, how dependent are these results on the initialization of A (a0 in this case). In particular, how robust are these results to variation in the a0 prior. From the prior used here, it appears as though the agent already has some knowledge of what to expect when it samples the risky option.

- Further to the issue of the prior – how robust are these results to the reward prior specified in matrix c? Can the authors justify the use of zero reward in all non rewarding states, but a -2 reward for a state that omits a reward? Do results depend on this initialization? More traditional models would encode the omission of a reward as zero – would punishment be encoded any differently (e.g. small foot shock instead of reward omission)?

- If the model is to be used for quantification of real datasets (i.e., fitting), then it would be necessary to verify that its parameters are recoverable / identifiable.

4) Other scenarios regarding the generality of the conclusions:

- The results outlined in Figure 10 show that the agent adapts to the consistent structure found in the environment, opting not to sample the informative cue once it has determined that the risky outcome will in fact be found reliably. If the environment were switched to offer a high reward 25% of the time once the agent was confident that it would get a reward 75% of the time would it resample the cue? That is, can the agent adapt its active inference to accommodate environments with low volatility periods?

- Behavior is often associated with some cost, which complicates the decision as to whether a cue should be sampled prior to reward pursuit. Does the model behave reasonably if there's a cost associated with sampling the cue?

[Editors' note: further revisions were requested prior to acceptance, as described below.]

Thank you for resubmitting your work entitled "Computational mechanisms of curiosity and goal-directed exploration" for further consideration at *eLife*. Your revised article has been favorably evaluated by Michael Frank (Senior Editor), a Reviewing Editor, and two reviewers.

As you will see the two reviewers had polarized views on the merits of the paper. The main concern is that some of the connections to empirical data are misleading, and that the paper is difficult to read. After discussion with them, and reading it myself, I think the latter concern partially reflects the fact that you thoroughly responded to original concerns during the triage process and those in the reviews, which may have made the case more compelling but also the paper became a little unwieldy (and part of that is to be expected for technical papers). Thus, I am asking that you again revise the paper, but this time mostly for clarity.

Essential revisions:

My first suggestion is thus that you take a full editing pass through the paper again (perhaps offering to an arms-length colleague for comments) to try to streamline the readability – I will leave it up to you how to do this. My second suggestion is to also try to tighten up the language in reference to the empirical data that reviewer 2 found to be 'vague'. Finally, it is great that you made all code accessible so that anyone can test the properties of the model themselves, but if there are additional ways you can expose key falsifiable predictions of the theory (beyond those that are postdictions of existing data) that would be useful.

*Reviewer #1:*

I feel that the authors sufficiently addressed most issues that we raised with the initial manuscript. They don't offer a demonstration that model parameters are recoverable, but I agree with their argument that perhaps this isn't necessary here. My only remaining issue is that, as a paper, it feels a bit long winded and somewhat fragmented – but this is more a matter of taste than it is of publishability.

*Reviewer #2:*

This is a revision of a paper that tries to explain the free energy framework and its significance for exploration and exploitation. The authors responded to the requests from the previous round of reviews to include a discussion of how their data can capture empirical observations by adding 7 (!) new figures, bringing the total to 21 (!) figures in the main text.

I am afraid that, although I do appreciate their effort, this revision does little to address my concerns. This, in my view, remains an engineering paper, concerned primarily on how to make a system that generates exploration and exploitation. It is not a paper that carefully considers – or even understands – empirical work, or that provides a model that makes *falsifiable* predictions or can be tested against *competing* models of empirical results.

The discussion of empirical results that the authors added is riddled with vague and meaningless – and often entirely wrong statements. Let me give just one example from subsection “Active inference and active learning in behavior”: "This is in line with previous work on curiosity and exploration, where attention and salience have been identified as central mechanisms that modulate curiosity.…" I am quite familiar with the papers that are cited and I am positive that they said very little about how "attention" and "salience" "modulate" curiosity. I wonder what the authors mean by this. Do they mean that the papers manipulated attention and tested if people were more curious when attention was diverted one way or another? (This was not the case). Did they mean that those papers manipulate salience and test how this produces curiosity? (Again, not the case.) And so on.

This is but one example – but the paper is *full* of such statements, to the extent that I find it, quite frankly, unreadable. I am very sorry to be so negative, especially since I appreciate the work the authors put in. I return to my original opinion – this time with even more conviction after this review – that this paper is simply not suitable for an empirical journal – it is a paper about computer science and engineering.

---

## [Author Response]

Essential revisions:Despite the strengths of the manuscript, the reviewers both had substantial reservations that would need to be addressed in a revision. The two reviewer points are consolidated below.1) The reviewers were both concerned about the lack of connection of the analysis to empirical data that could be used to constrain or falsify it. They felt that the simulations are simple enough that there should be some data out there that could corroborate their model's predictions, and that for a life sciences (instead of purely computational) journal this is important.- For example, the authors outline clear behavioral patterns that should be observed when an animal begins a simple t-maze task with risky options, or an experiment with a disambiguating cue. Are there existing data that could help support the model's claims here? Some evidence showing that rats preferentially sample from a risky option early, but switch to the low-risk option over time would certainly seem to be available in the corpus of animal behavior literature.

We agree with the referees that this analysis would substantially improve our manuscript. We now discuss specific behavioural (and neuronal, see below) predictions from our computational perspective in an additional subsection called “Behavioural and neuronal predictions”. The contribution of this section is twofold. First, we outline key behavioural and neuronal predictions that can be used to validate or falsify certain aspects of the model, such as the relationship between the (extrinsic) value of an option and its (intrinsic) informativeness. Second, we discuss these predictions in the light of existing behavioural and neuronal results in the animal literature (similar to Yu and Dayan, 2005, for instance). For example, we discuss behavioural evidence for an additive value of information in addition to reward (e.g., Vasconcelos, Monteiro and Kacelnik, 2015 in starlings) or a preference for sampling a risky option motivated by information gain, even if the safe option has higher expected utility (Smith, Beran and Young, 2017 in macaques). In particular, we discuss and simulate results from Blanchard, Hayden and Bromberg-Martin, (2015), showing evidence for a supra-additive effect of ‘information’ on the value of an option as a function of its (extrinsic) reward. Importantly, despite using a ‘rat’ as an example agent, the model-based predictions reported here are not restricted to rodents. Consequently, we discuss these predictions based on different examples from the entire animal literature.

We would be happy to include and discuss any additional results or references that we might have missed during our literature review.

In so doing, the reviewers also agreed that it would be important to compare your model to some alternatives, even if not doing formal model comparison with quantitative fitting, but at least qualitatively.

We agree that this is a useful extension to our manuscript. To address this, we have included an additional subsection called “Relationship to other computational models of exploration”, where we discuss some of the most prominent existing computational frameworks for exploration in reinforcement learning and deep neural networks, such as Thompson sampling or upper confidence bounds (similar to Gershman, 2018).

2) They also both agreed that there was very little connection to neural mechanisms despite various work that has been conducted in this domain. One reviewer noted "The decision to sample a cue to infer a hidden state involves cost-benefit tradeoffs relying on specific neural substrates but these are not mentioned at all. It also involves active sensing behaviors – such as whisking, about which a lot is known – but there is no real mention of any active sensing literature. The simulations in Figure 3 predict clear-cut risk aversion but there is no discussion of the fact that animals show mixtures of risk seeking and risk aversion when tested in these conditions. Active learning and active inference are associated with different neural substrates and behaviors but this is never mentioned when discussing the two mechanisms."

We agree that this discussion provides an important and useful addition to our manuscript. We now outline specific neuronal predictions based on active inference and active learning in the subsection “Behavioural and neuronal predictions” alongside key behavioural implications. We also assess the evidence for these predictions in relation to reported results on the neuronal implementation of active inference and active learning (e.g. Blanchard, Hayden and Bromberg-Martin, 2015 in monkeys; Ligneul, Mermillod and Morisseau, 2018 in humans and as discussed in Gottlieb, Oudeyer, Lopes and Baranes, 2013; Kidd and Hayden, 2015) and their relation to the active sensing literature (e.g., Yang, Lengyel and Wolpert, 2016; Yang, Wolpert and Lengyel, 2018). The model-based predictions described in our paper are also strongly aligned with identity (information) prediction errors that have recently been reported in rodents (Takahashi et al., 2017), which we now discuss in more detail. As above, given that our computational predictions are not limited to rodents, we discuss different examples from the animal literature that illustrate the neuronal implementation of active inference and active learning.

The issue of risk seeking and risk avoiding behaviour and its relation to ambiguity reduction is a very important one. We now discuss this in more detail in the subsection “Effects of other model parameters’”, please see below). Risk preferences are determined by an agent’s preferences over outcomes (c-vector), which can induce risk seeking or risk avoiding behaviour in agents.

3) Technically, there were also a few important concerns that would have to be addressed:- One concern was the flexibility of the free-energy framework, and how sensitive/robust it might be with respect to initial conditions and priors. For example, the agent in the first simulation takes an initialization as specified in Figure 2. Given that the agent is tasked with updating A, how dependent are these results on the initialization of A (a0 in this case). In particular, how robust are these results to variation in the a0 prior. From the prior used here, it appears as though the agent already has some knowledge of what to expect when it samples the risky option.- Further to the issue of the prior – how robust are these results to the reward prior specified in matrix c? Can the authors justify the use of zero reward in all non rewarding states, but a -2 reward for a state that omits a reward? Do results depend on this initialization? More traditional models would encode the omission of a reward as zero – would punishment be encoded any differently (e.g. small foot shock instead of reward omission)?

We agree that these are important issues to discuss. As we briefly mention in the manuscript, predicted behaviour also depends on other parameters in the model, including an agent’s prior on the observation model (a0) and preferences over outcomes (cost/utility function, c-vector). In the subsection “Effects of other model parameters” we now discuss the effects of these parameters (all available at https://github.com/schwartenbeckph/Mechanisms_Exploration_Paper).

We illustrate that the current specification of the prior on an agent’s observation model as reported in the manuscript implies high uncertainty about obtaining a reward in the risky option, by assigning an imprecise (i.e., low concentration parameters) uniform prior over the reward statistics in the risky option. Changing these priors affects an agent’s expectations about obtaining a reward in the risky option, either by inducing more optimistic or pessimistic expectations about the reward statistics, or by making the uniform prior more precise. We will illustrate the effects of such priors on resultant behaviour. The only other knowledge that this prior reflects is basic task-specific information, such as the fact that there will be no high reward in the safe option or no reward at all in the starting position. It is possible to assign prior uncertainty to the mappings in the safe option and starting position as well, which would associate these states with information gain, as we briefly illustrate in the manuscript ().

Second, we now illustrate the effects of an agent’s prior preferences on behaviour, including simulations where the omission of a reward is parameterised with a value of zero (as in Friston et al., 2015 or Schwartenbeck and Friston, 2016, for instance, see subsection “Prior preferences over outcomes determine risk preferences and the cost of information”). This does not affect the general behaviour or claims of the paper. Further, we also discuss the role of these priors in terms of their role as a cost or (negative) utility function. In the simulations presented in this manuscript, both rewards and punishments are accounted for in the same preference function (c-vector) and ‘punishment’ would simply have a more negative value than the omission of a reward (see subsection “Prior preferences over outcomes determine risk preferences and the cost of information”). Because prior preferences can also be understood as (negative) costs, visiting the cue before sampling the most valuable option is costly and needs to be justified by the information gain imparted by the cue (please see below).

- If the model is to be used for quantification of real datasets (i.e., fitting), then it would be necessary to verify that its parameters are recoverable / identifiable.

We feel that the point of our paper is a more theoretical and general one similar to recent discussions of theoretical accounts of exploration (e.g., Gershman, 2017; 2018), which also do not mention the specifics of model fitting procedures or parameter recovery. We believe that an additional subsection on model fitting and parameter recovery would be beyond the scope of this manuscript. Model fitting procedures and tests for parameter recovery are illustrated in detail in a recent tutorial on active inference and active learning (‘practical_II.m’ at https://github.com/schwartenbeckph/CPC_ActiveInference20182, also discussed in Schwartenbeck and Friston, 2016, for instance, which we now mention in subsection “Behavioural and neuronal predictions” and in our ‘code availability’ statement).

However, if the reviewers believe that including a subsection on model fitting and parameter recovery would be necessary, then we would be happy to add this to the manuscript or a supplement.

4) Other scenarios regarding the generality of the conclusions:- The results outlined in figure 10 show that the agent adapts to the consistent structure found in the environment, opting not to sample the informative cue once it has determined that the risky outcome will in fact be found reliably. If the environment were switched to offer a high reward 25% of the time once the agent was confident that it would get a reward 75% of the time would it resample the cue? That is, can the agent adapt its active inference to accommodate environments with low volatility periods?

This is an important observation that speaks to the implementation of active inference and learning in problems like reversal learning paradigms. We now discuss this in the final part of the subsection “Effects of other model parameters”. We show that active inference can accommodate reversals and dynamic changes in the environment. Active inference predicts that an agent will start to sample the cue again if its uncertainty about the current context increases following observations that indicate a change in task structure. We also show that this behaviour depends on prior expectations about the (stability of the) task structure.

- Behavior is often associated with some cost, which complicates the decision as to whether a cue should be sampled prior to reward pursuit. Does the model behave reasonably if there's a cost associated with sampling the cue?

This point speaks to the definition of the c-vector as the agent’s cost function or (negative) expected utilities. In fact, in the simulations that we present in this manuscript there is a cost for sampling the cue, because the cue is less valuable than sampling the safe (or high reward risky) option. That is the reason why the agent chooses to sample the safe or risky option right away once the ambiguity about the risk probabilities is resolved – even though it could still sample the cue first, because every trial consists of two time-steps. This is because sampling the cue has higher cost compared to the safe option (or risky option if it is associated with a reward probability of more than 0.5). One could also make the cue costlier by assigning it with a lower value in the c-vector, which would allocate a higher cost for visiting the cue compared to staying at the starting position, for instance. In general, in any situation where the safe (and high reward risky) option has lower cost (higher utility) than the cue, the value of information has to exceed the cost of visiting the cue to motivate the agent to explore the cue first at the beginning of a trial.

We now illustrate this sort of behaviour (subsection “Prior preferences over outcomes determine risk preferences and the cost of information”).

[Editors' note: further revisions were requested prior to acceptance, as described below.]

As suggested, we have thoroughly revised our manuscript with the aim of making it more readable and improving its clarity.

In particular, we have made the following changes:

We have now moved two subsections of the main manuscript to an appendix, namely the subsection “Effects of other model parameters” and subsection “‘Relationship to other computational approaches to exploration’”. While we believe that these are important sections, they are less critical for conveying the general claims of the paper, and we hope that the main manuscript has become more coherent that way.

Consequently, we have also reduced the number of figures from 21 to 13 in the main text.

We have tried to clarify the language throughout the paper (changes are marked in yellow), with particular emphasis on the issues raised by referee 2.

As suggested, we have re-written and clarified our subsection on open questions and empirical predictions, following our discussion of empirical evidence for active inference and active learning. In behaviour, it is often possible to find a prior parameterisation that explains individual behaviour that seems suboptimal or deviant from model predictions. Nevertheless, we discuss specific predictions from our model that arise from the (supra) additivity assumption of information and reward and the discounting of future information. Neuronally, we discuss predictions and open questions about the neuronal encoding of information as predicted by our framework, in particular (empirically testable) predictions about dopaminergic responses to information and the interplay of a factorised (prefrontal cortex) and a conjunctive (dopamine neurons) value code.

We have now split up our description of the model into a more conceptual description in our Results section and a more technical description in our Materials and methodsmethods section at the end of the manuscript.

We have also tried to make the manuscript less “fragmented” (pointed out by referee 1) by describing how the different sections connect to each other at the beginning and end of the individual sections.

We hope that this is what you had in mind and that the manuscript has become more readable now. We would be happy to treat any of these aspects in more detail if you feel that our treatment of any of the raised issues was not sufficient.